# Encoding sensory and motor patterns as time-invariant trajectories in recurrent neural networks

**Vishwa Goudar[1]\*, Dean V Buonomano[1,2,3]\***

[1]Departments of Neurobiology, University of California, Los Angeles, Los Angeles, United States; [2]Integrative Center for Learning and Memory, University of California, Los Angeles, Los Angeles, United States; [3]Departments of Psychology, University of California, Los Angeles, Los Angeles, United States

**Abstract** Much of the information the brain processes and stores is temporal in nature—a spoken word or a handwritten signature, for example, is defined by how it unfolds in time. However, it remains unclear how neural circuits encode complex time-varying patterns. We show that by tuning the weights of a recurrent neural network (RNN), it can recognize and then transcribe spoken digits. The model elucidates how neural dynamics in cortical networks may resolve three fundamental challenges: first, encode multiple time-varying sensory *and* motor patterns as stable neural trajectories; second, generalize across relevant spatial features; third, identify the same stimuli played at different speeds—we show that this temporal invariance emerges because the recurrent dynamics generate neural trajectories with appropriately modulated angular velocities. Together our results generate testable predictions as to how recurrent networks may use different mechanisms to generalize across the relevant spatial and temporal features of complex time-varying stimuli.

DOI: https://doi.org/10.7554/eLife.31134.001

**\*For correspondence:**
vishwa.goudar@gmail.com (VG);
dbuono@ucla.edu (DVB)

**Competing interests:** The authors declare that no competing interests exist.

## Introduction

Many, if not most, of the tasks the brain performs are inherently temporal in nature: from recognizing and generating complex spatiotemporal patterns—such as a phoneme sequence that composes a word—to creating temporal expectations of when an event will occur (*Mauk and Buonomano, 2004*; *Nobre et al., 2007*; *Ivry and Schlerf, 2008*; *Merchant et al., 2013*; *Hopfield, 2015*). In contrast to the representation of an object in a static image, such as a picture of a face, robust decoding and encoding of a time-varying patterns must rely on the spatiotemporal dependencies inherent to the pattern. How does the brain accomplish this? One highly influential theory in neuroscience holds that information is stored as fixed-point attractors that emerge in the dynamic activity of the brain's recurrently connected circuits (*Hopfield, 1982*; *Hopfield and Tank, 1986*; *Amit and Brunel, 1997*; *Wang, 2001*). Two limitations of this framework are: (1) It fails to capture the temporal aspect of stored information, thus forcing many computational models to 'spatialize' time—that is, they treat the temporal component of time-varying patterns as additional spatial dimensions (*Rabiner, 1989*; *Waibel et al., 1989*; *Elman, 1990*; *Hinton et al., 2012*; *Mnih et al., 2015*); (2) it does not capture a fundamental feature of how the brain processes temporal information: temporal invariance. For example, humans readily recognize temporally warped—compressed or dilated—speech or music.

While the mechanisms that underlie the brain's ability to perform a broad range of spatiotemporal tasks in the sensory and motor domains are not known, there is mounting theoretical and experimental evidence that our ability to tell time on the sub-second scale, and represent time-varying patterns, relies on the inherent *continuous* dynamics, and computational potential, of recurrent

neural networks (*Rabinovich et al., 2008*; *Buonomano and Maass, 2009*; *Mante et al., 2013*; *Crowe et al., 2014*; *Carnevale et al., 2015*; *Li et al., 2016*). This framework alleviates the restrictions imposed by traditional discrete-time or fixed-point attractor models, by relying on the recurrent connections to implicitly maintain an ongoing memory of the pattern. Indeed, recent computational studies have established that by tuning the weights within recurrent neural network (RNN) models, it is possible to *robustly* store and generate complex time-varying motor patterns (*Laje and Buonomano, 2013*; *Abbott et al., 2016*; *Rajan et al., 2016*). What remains unknown is whether the same approach can be used to robustly discriminate time-varying sensory patterns. Indeed, this poses a challenging problem because in order to effectively process spatiotemporal stimuli the dynamics of an RNN must be sensitive to the relevant spatial and temporal features of the sensory stimuli, while being able to generalize across their natural spatial and temporal variations.

Neuroscientists have typically distinguished between sensory and motor areas; but it is well established that activity in sensory areas is strongly influenced by motor behavior, and that sensory stimuli can modulate activity in motor areas—furthermore, some brain areas are characterized as being sensorimotor (*Doupe and Kuhl, 1999*; *Ayaz et al., 2013*; *Chang et al., 2013*; *Schneider and Mooney, 2015*; *Cheung et al., 2016*). Computationally, sensory and motor processing are understood to have disparate requirements—during sensory processing, network dynamics should primarily be driven by the sensory inputs; in contrast, during motor processing neural dynamics should be autonomous and driven primarily by recurrent interactions. It remains unclear how a single network could accomplish both tasks, and couple them when necessary. Indeed, to date, no previous models have shown that the same network can satisfy the requirements for both sensory and motor processing. Here we show that the same RNN can reliably function in both a sensory and motor regime. Specifically the same RNN can convert complex time-varying sensory patterns into motor patterns—thus performing a transcription task in which spoken digits are identified and read out as 'handwritten' digits.

In the temporal domain, understanding how the brain recognizes temporally compressed or dilated patterns represents a long-standing challenge. This temporal invariance is particularly evident in our ability to recognize temporally warped speech or music (*Miller et al., 1984*; *Sebastián-Gallés et al., 2000*). Our results suggest that one advantage of storing time-varying patterns as neural trajectories within RNNs is that, under the appropriate conditions, this strategy naturally accounts for temporal invariance.

## Results

We first asked if a single RNN could perform a complex sensory-motor task: transcribing spoken digits into handwritten digits. A *continuous-time* firing-rate RNN with randomly assigned sparse recurrent connections was used (*Sompolinsky et al., 1988*). The strengths of these recurrent connections were initialized to be relatively strong; thus, before training the network was in a so-called high-gain regime ($g$ = 1.6) that is characterized by self-perpetuating and chaotic activity (Materials and Methods). The transcription task is divided into a sensory and a motor epoch. During the sensory epoch, the RNN is presented with a spoken digit, and over the ensuing motor epoch the RNN drives an output pattern transcribing the presented digit via three motor outputs $x$, $y$, and $z$—activity in the $x$ and $y$ units determines the 2D coordinates of a 'pen on paper', while $z$ determines if the 'pen' is in contact with the 'paper' or not. *Figure 1A* illustrates the network architecture and transcription task for the digit '2'. Successful performance of this transcription task requires that the RNN: 1) encode spoken digits with a set of neural trajectories in high-dimensional phase space; 2) autonomously generate digit-specific trajectories during the motor epoch, in order to drive the digit-specific output patterns; and most importantly 3) generate trajectories that are stable so they may encode each digit's sensory pattern, sensory-motor transition, and motor pattern in a manner that is invariant not just to background noise but also to spatiotemporal variations of the spoken digits, including temporal warping.

We attempted to satisfy these three conditions by training the RNN using a supervised learning rule, in which the recurrent units were trained to robustly reproduce their 'innate' patterns of activity (*Laje and Buonomano, 2013*)—that is, those generated in the untrained network—using the recursive-least-squares learning rule (*Haykin, 2002*; *Sussillo and Abbott, 2009*). In essence, the RNN is

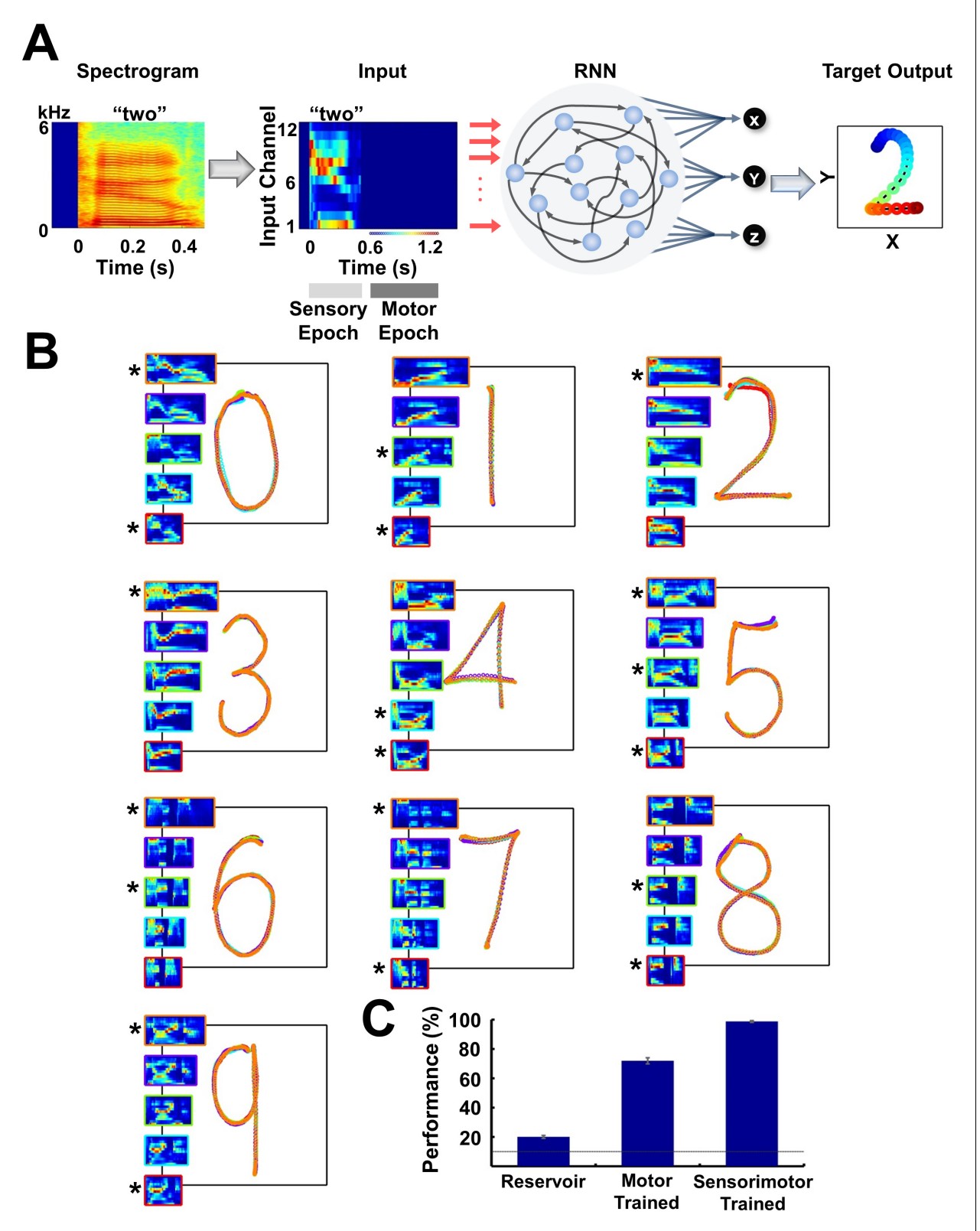

**Figure 1.** Trained RNNs perform a sensorimotor spoken-to-handwritten digit transcription task (**A**) Transcription task. The spectrogram of a spoken digit, e.g. 'two', is transformed to a 12-channel cochleogram that serves as the continuous-time input to a RNN during the sensory epoch of each trial. During the motor epoch, the output units must transform the high-dimensional RNN activity into a low-dimensional 'handwritten' motor pattern corresponding to the spoken digit (the z output unit indicates whether the pen is in contact with the 'paper'). The colors of the output pattern (right

*Figure 1 continued on next page*

*Figure 1 continued*

panel) represent time (as defined in the 'Input' panel). (B) Overlaid outputs of a trained RNN (*N* = 4000) for five sample utterances of each of 10 digits. For all digits, each output pattern is color-coded to the bounding box of the corresponding cochleogram (inset). Sample utterances shown are a mix of trained (*) and novel utterances, and span the range of utterance durations in the dataset. (C) Transcription performance of three different types of RNNs on novel utterances. Performance was based on images of the output as classified by a deep CNN handwritten digit classifier. The control groups include untrained RNNs ('reservoir') and RNNs trained only during the motor epoch (i.e., just to reproduce the handwritten patterns; see *Figure 1—figure supplement 1*). Output unit training was performed identically for all networks. Bars represent mean values over three replications, and error bars indicate standard errors of the mean. Line indicates chance performance (10%). The RNNs were trained on 90 utterances (10 digits, three subjects, three utterances per subject per digit). They were then tested on 410 novel utterances (across five speakers, including two novel speakers), with 10 test trials per utterance. $I_0$ was set to 0.5 during network training (if applicable), and to 0.05 during output training and testing.
DOI: https://doi.org/10.7554/eLife.31134.002

The following figure supplement is available for figure 1:

**Figure supplement 1.** Transcription performance of an RNN trained only during the motor epoch.
DOI: https://doi.org/10.7554/eLife.31134.003

trained to reproduce one digit-specific 'innate' trajectory in response to all training utterances of a given digit. For example, the pattern of activity produced in response to a 'template' utterance of a given digit from the beginning of the sensory epoch to the end of the motor epoch is taken as the innate trajectory; the network is then trained to reproduce this trajectory in response to other utterances of the same digit. Furthermore, it is trained to do so regardless of the initial state of the network's units, and in the presence of continuous background noise (Materials and Methods). Only after training the RNN (*recurrent training*), are the output units trained to generate the handwriting patterns representing each of the digits during the motor epoch (*output training*)—this separation of the recurrent and output training phases, while not necessary, allows for rapid training and retraining of the outputs to produce arbitrary motor patterns, without retraining the recurrent weights. Notably, the temporal separation of sensory and motor learning is observed in some forms of sensorimotor learning (*Doupe and Kuhl, 1999*). The output training phase is performed using standard supervised methods (Materials and Methods). *Figure 1B* shows the outputs of a trained network cross-tested on ten digits (0–9) across five speakers. Following training, the network successfully transcribes the utterances used during training (marked by asterisks). More importantly, performance generalizes to novel utterances and speakers in the dataset despite significant variations in the duration and spatiotemporal structure across utterances (*Video 1*).

To quantify performance, we need an objective measure of the quality of the motor output for each test utterance. This itself represents a classification task, wherein images of RNN-generated transcriptions must be assigned to one of the ten possible digits. Rather than use a human-based performance measure, we used a standard deep convolutional neural network (CNN) (*LeCun et al., 2015*) to rate the performance of trained RNNs (Materials and Methods)—i.e., the CNN was used to determine if each 'handwritten' digit was correct or not. Performance of trained RNNs was 98.7% on a test set of 410 novel stimuli (*Figure 1C*). Since untrained RNNs ('reservoir networks') are in and of themselves capable of performing many interesting computations (*Maass et al., 2002*; *Jaeger and Haas, 2004*; *Lukoševičius and Jaeger, 2009*), it is important to determine how much of this performance is dependent on the training of the

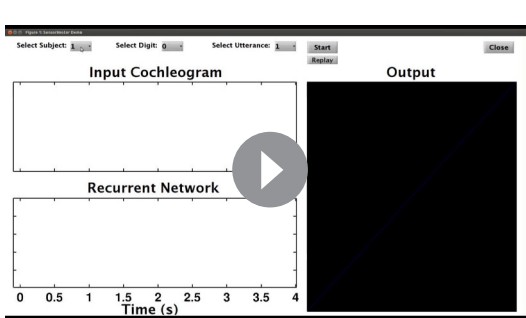

**Video 1.** A trained RNN performs the sensorimotor spoken-to-handwritten digit transcription task on novel utterances. A trained RNN (N = 4000) and its output units perform the transcription task on five novel utterances (five different speakers). The last two utterances illustrate RNN performance on same-digit utterances of different durations (i.e. temporal invariance).Top Left: Input to the RNN (with audio). Moving gray bar indicates the current time step. Bottom Left: Evolution of RNN activity in a subset of the network (100 units). Right: Evolution of the output. The location of the pen is imprinted in red when the z co-ordinate is greater than 0.5, and plotted in gray otherwise.
DOI: https://doi.org/10.7554/eLife.31134.004

RNNs. We thus examined a control group wherein the three output units are trained as they were for the trained RNNs, but the RNN itself was not trained (the 'reservoir'). The performance was very poor (20%), in large part because during the motor epoch a reservoir RNN is operating in an autonomous mode that is chaotic—making it difficult for the output units to learn to produce the target output patterns. We also examined the effects of training an RNN only during the motor epoch, resulting in a performance of 71%. This significant improvement over the reservoir networks is a consequence of the fact that a network driven by external inputs can encode sensory stimuli despite a lack of sensory epoch training, due to the intrinsic ability of RNNs to encode sensory stimuli and the stabilizing influence of the external inputs (see below). Yet, despite the improvement in performance, some digits were almost always misclassified (*Figure 1—figure supplement 1*). The difference in performance between the trained RNNs (sensory and motor training) and the exclusively motor trained RNNs, confirms the importance of tuning the recurrent weights to the sensory discrimination component of the task (see below).

## Stability of the neural trajectories

RNNs operating in high-gain regimes are prone to chaotic behavior (*Sompolinsky et al., 1988*; *Rajan et al., 2010a*) because the strong recurrent feedback of these networks rapidly amplifies any noise or perturbations. It is thus critical to demonstrate that the above performance is robust to background noise and perturbations during the sensory and motor epochs. The distinction between epochs is critical since the RNN is operating in fundamentally different regimes during the sensory and motor epochs. During the sensory epoch the recurrently generated internal dynamics are partially suppressed (or 'clamped') as a result of the external input, yet during the motor epoch all activity is internally generated. To examine the stability of these sensory and motor 'object' representations, we briefly perturbed the internal dynamics of the RNN during either the sensory or motor epochs. Sensory epoch perturbations were introduced half way through the presentation of each utterance, while motor epoch perturbations were introduced at the 10% mark of the motor epoch. *Figure 2A* shows the activity of a hundred units from a trained RNN (and the resulting output), when it is presented with the digit 'three' and strongly perturbed (amplitude = 2) in the motor epoch. The resulting transcription briefly deviates from an unperturbed one (grey backdrop), but recovers mid-voyage. The sensory epoch is less sensitive to perturbations—as would be expected because the presence of the external input serves as a stabilizing influence (*Rajan et al., 2010a*). During the motor epoch the RNN is operating autonomously as a 'dynamic attractor', that is, once bumped off its trajectory it maintains a memory of its current voyage and is able to return to the original trajectory (*Laje and Buonomano, 2013*) (*Figure 2B*). Performance measurements using the CNN classifier reveal a graceful degradation across a wide range of perturbation magnitudes, and confirm the superior robustness of the sensory epoch (*Figure 2C*).

## RNN training sculpts network dynamics to enhance discrimination

How does tuning the recurrent weights allow the RNN to stably encode both sensory and motor information in neural trajectories, despite considerable spatiotemporal differences between digit utterances (*Figure 1B*, insets)? Firing rate traces of sample units in a trained RNN show more similar patterns of activity in response to different utterances of the same digit, when compared to a reservoir RNN (*Figure 3A–B*). To better examine the structure of the population representations, we can visualize and compare the neural trajectories in response to multiple utterances (learned and novel) of the same digit, during the sensory and motor epochs, in principal component analysis (PCA) subspace. The trajectories produced in the untrained network by utterances of the digits 'six' and 'eight', during the sensory epoch, occupied a fairly large abutting volume of PCA subspace (*Figure 3C*); and during the motor epoch the trajectories were highly variable—the network strongly and perpetually amplifies the differences between utterances, because it is in a chaotic regime. In contrast, in the trained RNN, sensory epoch trajectories of each digit were restricted to a narrower volume, and better separated from the volume representing the other digit (*Figure 3D*). During the motor epoch, the trajectories for different utterances of the same digit were constrained to a much narrower tube—reflecting the dynamic attractor—and better separated from the motor trajectories of the other digit. The set of all within-digit trajectories can be thought of as populating a hypertube that delimits the volume of phase space traversed by the digit. During the sensory epoch, this

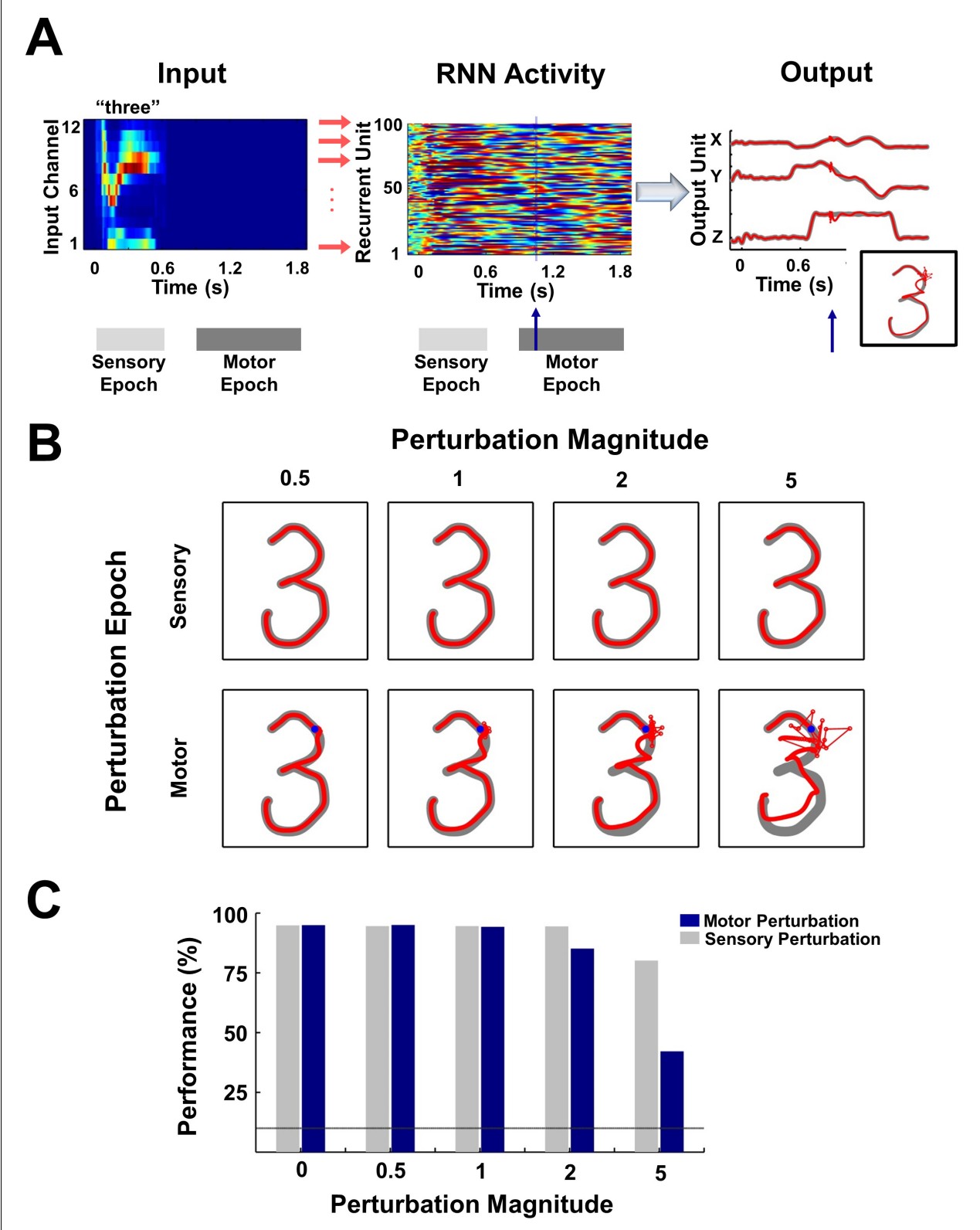

**Figure 2.** Digit transcription is robust to perturbations during the sensory and motor epochs. (**A**) Schematic of a perturbation experiment. The motor trajectory of a trained RNN (N = 2100; 100 sample units shown) for the spoken digit 'three', is perturbed with a 25 ms pulse (amplitude = 2). The pulse (blue arrow) causes a disruption of the network trajectory, but the trajectory quickly recovers and returns to the dynamic attractor, as is evident from the plots on the right comparing the output unit values and the transcribed pattern for a trial with (red forefront) and without (gray backdrop) the

*Figure 2 continued on next page*

*Figure 2 continued*

perturbation. (B) Sample motor patterns generated by the network in response to perturbations of increasing magnitude, applied either during the sensory or motor epochs. Sensory epoch perturbations were applied halfway into the epoch, while motor epoch perturbations were applied at the 10% mark of the epoch (indicated by a blue dot). (C) Impact of perturbations on transcription performance (measured by the deep CNN classifier) for test utterances. Bars represent mean performance over ten trials, with a different (randomly selected) perturbation pulse applied at each trial. Line indicates chance performance. The performance measures establish that the encoding trajectories are stable to background noise perturbations, with transcription performance degrading gracefully as the perturbation magnitude increases. Furthermore, at all perturbation magnitudes, the sensory encodings are more robust than their motor counterparts due to the suppressive effects of the sensory input. The network was trained on 30 utterances (3 utterances of each digit by one subject) and tested on 70 (7 utterances of each digit by one subject). $I_0$ was set to 0.25 during network training, to 0.01 during output training, and to 0 during testing except for the duration of the perturbation pulse.

DOI: https://doi.org/10.7554/eLife.31134.005

hypertube of trajectories represents a memory of a family of related spatiotemporal objects: the spoken digit. During the motor epoch the network is autonomous, and the hypertube of all within-digit trajectories narrows to a dynamic attractor that represents the 'motor memory' of the corresponding handwritten digit.

It is well established that cortical circuits undergo experience-dependent plasticity—a process that seems to result in the optimization or specialization of those circuits to the tasks the animal is exposed to (*Buonomano and Merzenich, 1998*; *Crist et al., 2001*; *Feldman and Brecht, 2005*; *Karmarkar and Dan, 2006*). The above results establish that training the RNN does improve discrimination performance—the recurrent weights are tuned to the task at hand—but leaves open the question of how exactly this is accomplished. To answer this question, we asked if the Euclidean distances between trajectories in response to different utterances of the same digit (within-digit distance), and utterances of different digits (between-digit distance), were significantly altered in comparison to the reservoir (untrained) RNN. Training significantly decreases the mean within-digit distances during the sensory epoch (*Figure 4A*). Importantly, in doing so, it does not diminish the large separations between trajectories of different digits. The same effect, much enhanced, is observed during the motor epoch, a consequence of the formation of dynamic attractors. Successful formation of these attractors is strongly influenced by an unambiguous and stable sensory epoch-motor epoch transition of the network dynamics. Such a transition relies on two factors: (i) a low within-digit separation at the end of the sensory epoch trajectory, ensuring initial conditions at the beginning of the motor epoch that lie within the initial basin of attraction of the motor pattern-encoding dynamic attractor; (ii) a between-digit separation that is substantially larger than the within-digit separation of sensory epoch trajectories, particularly at the end of the sensory epoch, to ensure robust and divergent sensory-epoch-motor epoch transitions for different digits.

The finding that training sculpts the dynamics of the RNN during the sensory epoch is also a critical one because it shows that even though RNN activity is governed in part by the external input, the internal connections are critical: tuning them effectively improves the interaction between the sensory input and the internally generated dynamics. This improvement is expressed as a collapsing of the family of trajectories representing a digit into a narrower hypertube. These results follow as a direct consequence of the supervised recurrent training paradigm (Materials and Methods)—a single innate trajectory serves as a common target for different training utterances of a digit, thereby inducing changes in the network's recurrent dynamics that are necessary to encode the different utterances along similar neural trajectories; However, different digits are encoded in rapidly divergent trajectories because the target trajectories for each digit are generated by an untrained chaotic network. Further evidence that training dramatically influences the weight matrix of the RNN is demonstrated by the change in its eigenspectrum (*Figure 4B*). Because the network is initialized with normally distributed weights with a gain of 1.6 (Materials and Methods), the eigenspectrum of the reservoir's weight matrix lies in a circle of radius approximately 1.6. Training results in a compression of those eigenvalues with real part larger than one, bringing the maximal real part of the eigenvalues closer to one. In a linear network, when all eigenvalues have a real part less than one, it implies that the network's activity will decay to zero when it operates autonomously; however our network is nonlinear and does not operate exclusively in the autonomous mode, so interpretations of the eigenspectrum of the weight matrix are not straightforward—nevertheless the compression of the

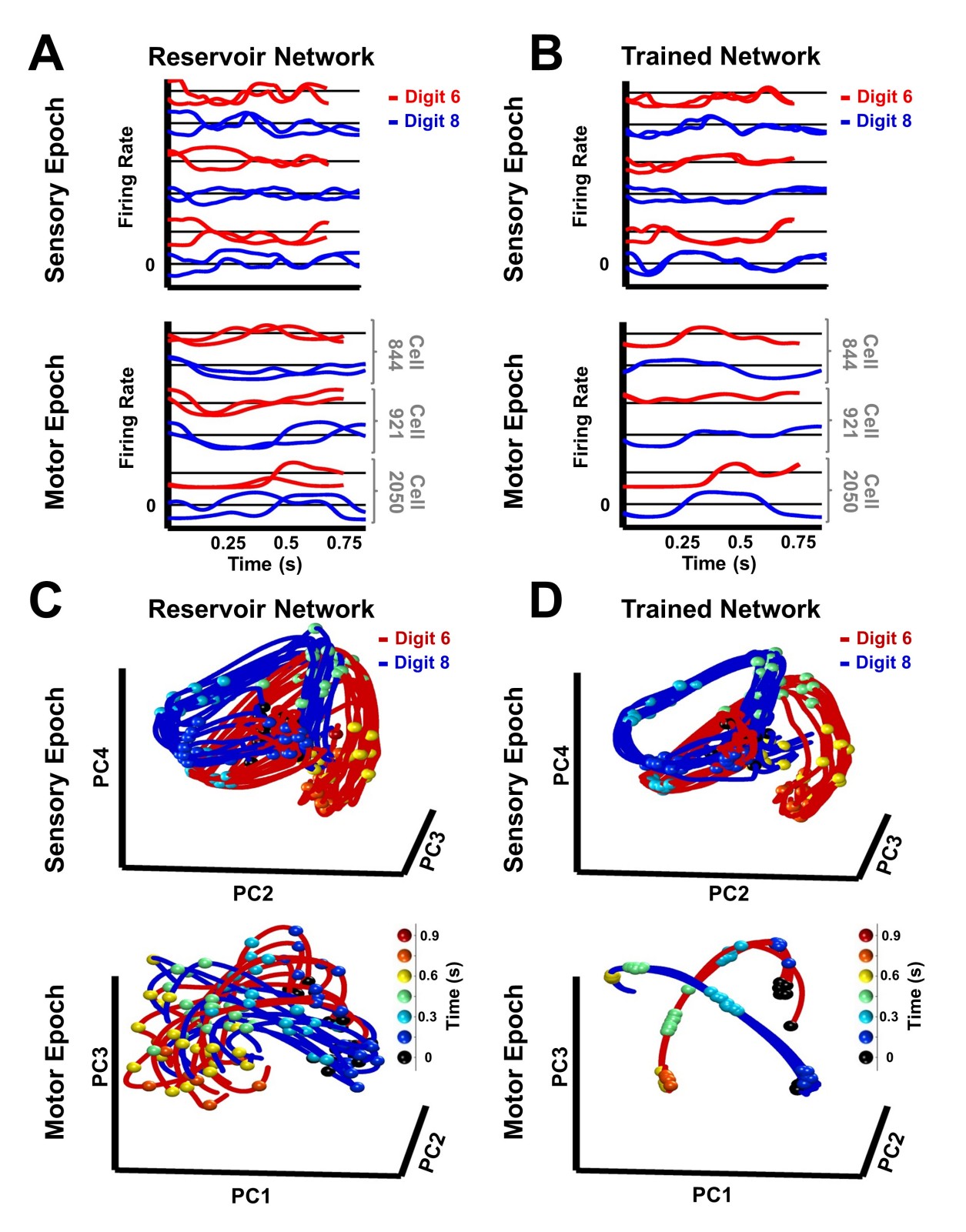

**Figure 3.** Trained RNNs generate convergent continuous neural trajectories in response to different instances of the same spatiotemporal object. (A–B) Neural activity patterns of three sample units of a reservoir (A) and trained (B) network, in response to a trained and a novel utterance each of the digits 'six' (red traces) and 'eight' (blue traces) during the sensory (top) and motor (bottom) epochs. Utterances of similar duration were chosen for each digit, to allow for a direct comparison, without temporal warping, of the corresponding pair of sensory epoch traces. (C–D) Projections in PCA space of the

*Figure 3 continued on next page*

*Figure 3 continued*

sensory and motor trajectories for 10 utterances each of the digits 'six' and 'eight' generated by the reservoir (C) and trained (D) networks. Colored spheres represent time intervals of 150 ms. Compared to the reservoir network, in the trained RNN the activity patterns in response to different utterances of the same digit are closer, and the trajectories in response to different digits are better separated. Both networks were composed of 2100 units. Network training was performed with 30 utterances (one subject, 10 digits, three utterances per digit). $I_0$ was set to 0.5 during network training, and to 0 while recording trajectories for the analysis.
DOI: https://doi.org/10.7554/eLife.31134.006

eigenspectrum is consistent with the increased stability of the network during the sensory and motor epochs (*Rajan and Abbott, 2006*; *Ostojic, 2014*).

## Balance between recurrent and input dynamics is crucial for discrimination

The above results provide insights as to how a single neural circuit can function in two seemingly distinct computational modes: sensory and motor. Sensory discrimination requires circuits to be highly responsive to external stimuli in order to categorize inputs into discrete classes. In contrast, motor tasks require autonomous generation of spatiotemporal patterns, and thus need the network dynamics to be somewhat resistant to external inputs. In the network described here, this balance depends on the recurrent weights and the magnitude of the external drive.

The recurrent weights must be strong—that is the network must be in a high-gain regime—for two reasons. First, network dynamics in high-gain regimes are high-dimensional (*Rajan et al., 2010b*), which naturally yields well-separated trajectories for different digits. A goal of training then, is to achieve recurrent suppression of within-digit separation, while retaining the large between-digit separation (*Figure 4*). Second, as noted earlier, a network operating in a high-gain regime is inherently capable of generating the self-perpetuating activity required in the motor epoch, which is then stabilized by the training procedure. Thus during the sensory epoch, the RNN is operating in a regime that is both sensitive to external input and influenced by strong internal recurrent dynamics. In the motor mode, the RNN operates autonomously, generating locally stable trajectories that serve as a high-dimensional 'engine' that drives arbitrary low dimensional output patterns.

The ability of the network to meaningfully process input patterns during the sensory epoch, so that they are easy to discriminate, also depends on the strength of the inputs. To better understand the impact of input amplitude on the encoding trajectories and their discriminability, before and after training, we parametrically varied the input amplitude and studied the resulting RNN dynamics during the sensory epoch. In the reservoir, both within and between-digit distances diminish as the input amplitude increases (*Figure 5A*, left panel), consistent with a previous study showing that strong inputs can progressively dominate and override the internal dynamics of an RNN (*Rajan et al., 2010a*). In the extreme, input-dominated regimes effectively void the recurrent weights and render training useless (*Figure 5A*, right panel). The poor discriminability at high input amplitudes is further confirmed by dimensionality measurements of sensory epoch trajectories (Materials and Methods), which mirror the low-dimensionality of the input cochleograms, regardless of training (*Figure 5B*). In contrast, low-input amplitude regimes are dominated by the RNN's internal dynamics. Reservoirs in this regime produce chaotic high-dimensional dynamics and are insufficiently sensitivity to external inputs; thus the within- and between-digit distances are similarly high (*Figure 5A*, left). Training strongly alters these dynamics, lowering its dimensionality (*Figure 5B*). However, as a consequence of poor input-sensitivity, these changes fail to improve discrimination—trained RNNs in this regime still sustain similar within and between-digit distances (*Figure 5B*, right). RNNs trained at intermediate input amplitudes (0.5, 5) are both sensitive to the input and able to discriminate between digits, because their sensory epoch trajectories are shaped both by the input and internal dynamics. It is thus critical that the input drive be strong enough to influence ongoing activity but not strong enough to 'erase' recent information encoded in the current trajectory.

## Mechanisms underlying spatial (Spectral) Generalization

The attracting dynamics of the network result in each digit being represented in a narrow hypertube. These representations must be robust across both spectral and temporal (changes in speed)

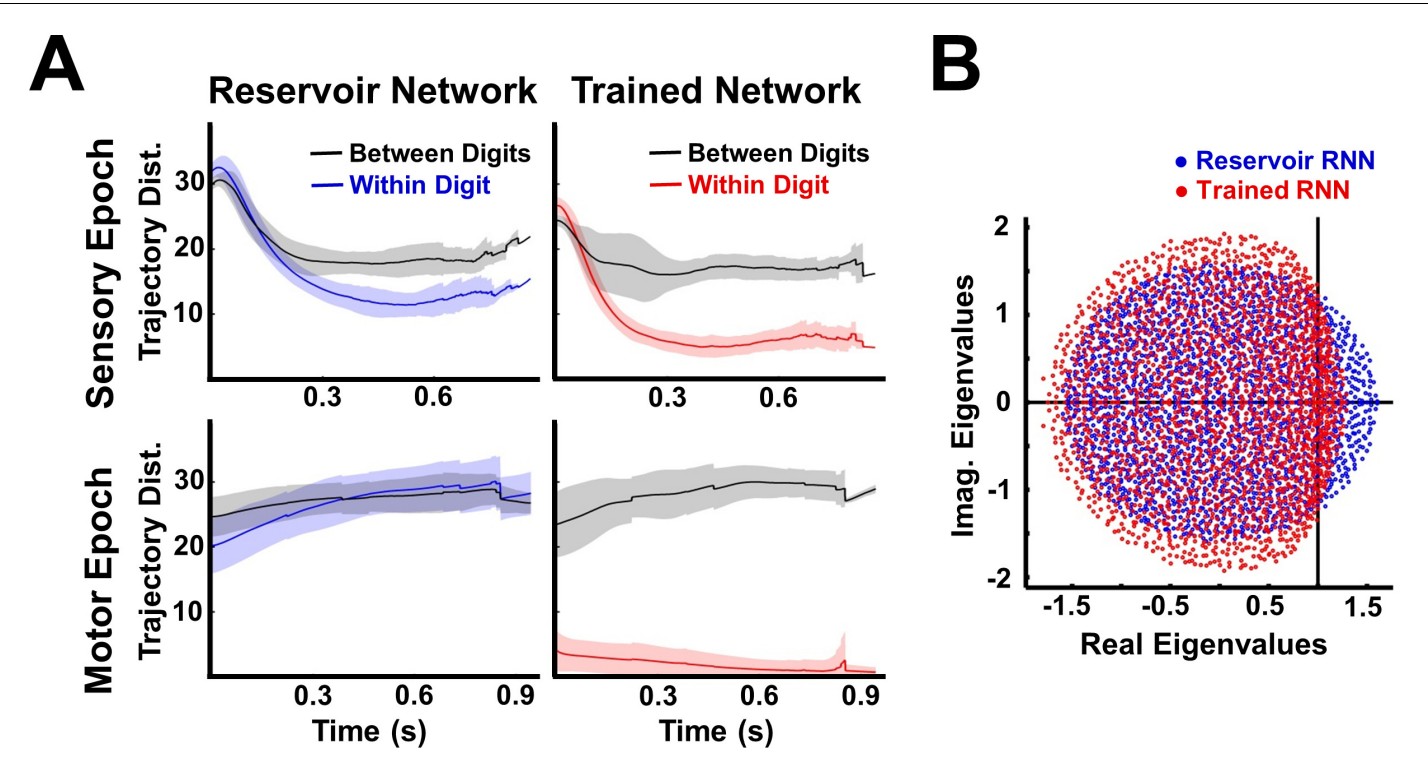

**Figure 4.** Trained RNNs encode both sensory and motor objects as well separated neural trajectories. (**A**) Euclidean distance between trajectories of the same digit (within-digit) versus those of different digits (between-digit). At each time step, the trajectory distances represent the mean and SD (shading) over pairs of one hundred utterances (one subject, 10 digits, 10 utterances per digit). During the sensory epoch, training brings trajectories for the same digit closer together, while maintaining a large separation between trajectories of different digits, thereby improving discriminability. A similar, but stronger, effect is observed during the motor epoch. (**B**) Comparison of the eigenspectrum of the recurrent weight matrix ($W^R$) in a reservoir and trained network. Both networks were composed of 2100 units. Network training was performed with 30 utterances (one subject, 10 digits, three utterances per digit). $I_0$ was set to 0.5 during network training, and to 0 while recording trajectories for the analysis.
DOI: https://doi.org/10.7554/eLife.31134.007

differences in the utterances of each digit. Next, we separately examine the mechanisms underlying spatial and temporal generalization. Variation in the spectral structure of digits—spectral noise (*Figure 6A*)—is qualitatively different from background noise. First, spectral noise exhibits time-dependent structured relationships with the input signal (they may be correlated, anti-correlated or tend to occur along specific directions during specific parts of the input). For example, the spectral differences between utterances of one phoneme within a digit will be different than those of another. Invariance then requires the network to isolate and integrate the signal while suppressing signal-dependent spectral noise. Second, spectral noise is a form of correlated noise. The noise in each input channel is simultaneously delivered to multiple recurrent units via common input projections. Moreover, the spectral noise in different channels, particularly neighboring ones, exhibit structured time-dependent relationships. Third, spectral noise is proportional in magnitude to the input amplitude, and therefore tends to be much stronger than the background noise that is typically injected during training and testing.

To better characterize how spectral noise invariance emerges during training, we measured network responses while naturally and artificially varying the structure of the spectral noise. Any confounding effects from temporal invariance were avoided by training and testing the network on temporally normalized utterances, by warping them to the duration of the template utterance for the respective digit. Responses to standard natural test utterances (those not presented during training) were contrasted with two types of artificially created utterances: those with shuffled spectral noise and orthogonal spectral noise (*Figure 6B*). Shuffled spectral noise was created by shuffling the PCA axes (basis) of the spectral noise (Materials and Methods). In other words, novel utterances

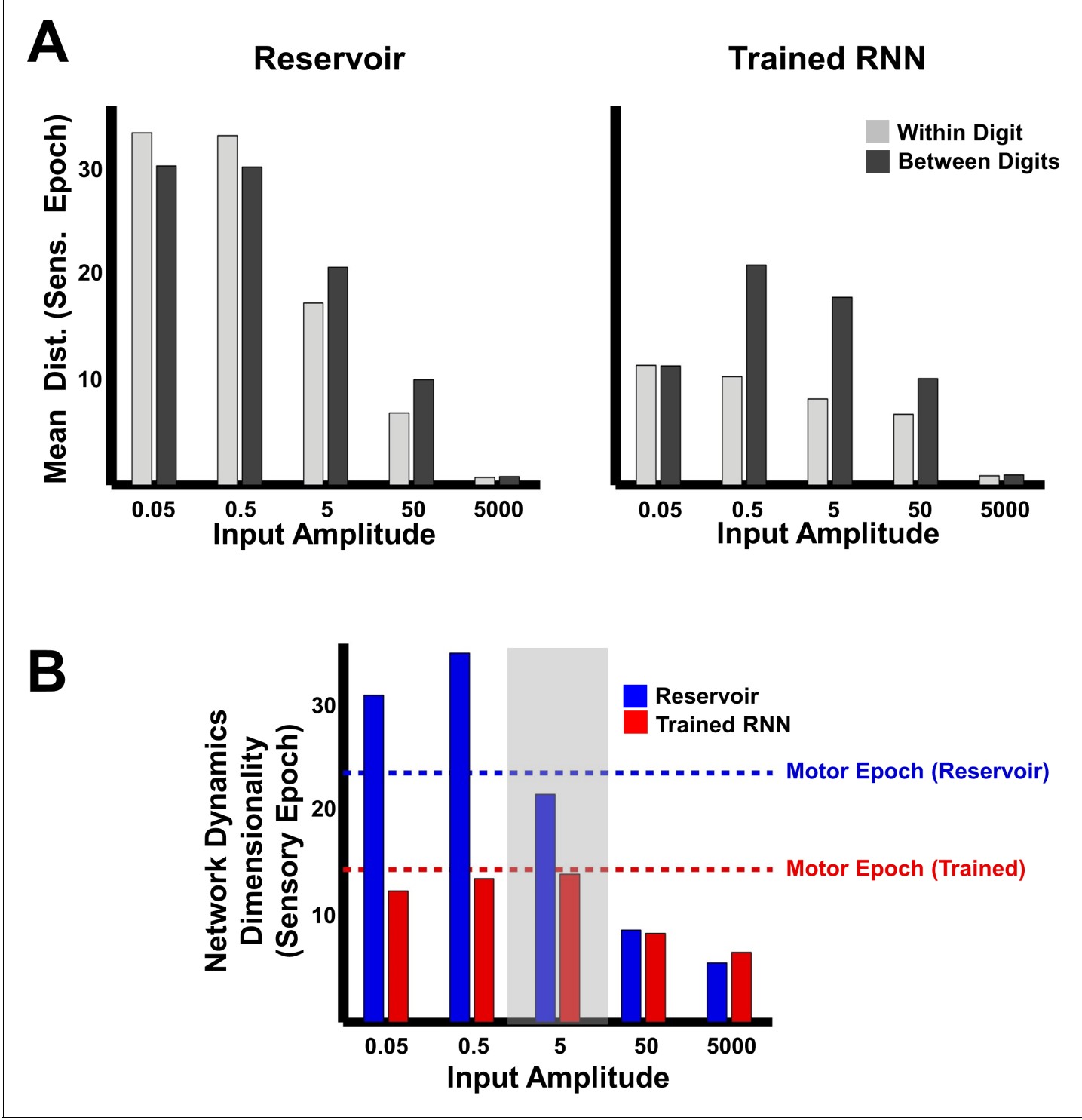

**Figure 5.** Trajectory separation in reservoir and trained RNNs as a function of input amplitude. (A) Comparison of mean within- and between-digit distances of the sensory epoch trajectories in reservoir and trained networks (N = 2100) at different input amplitudes. Bars represent mean of the time-averaged distances for all utterances of all digits (one subject, 10 digits, 10 utterances per digit). (B) Dimensionality of sensory epoch trajectories in the reservoir and trained networks at different input amplitudes. Simulations, including training, in all other figures were performed at an input amplitude of 5 (gray highlight). Dashed lines indicate the dimensionality of the motor epoch trajectories in the reservoir (blue) and trained (red) networks, when the input amplitude was 5. Network training was performed with 30 utterances (one subject, 10 digits, three utterances per digit). The networks were trained with $I_0$ set to 0.25, and trajectories were recorded with $I_0$ set to 0.

DOI: https://doi.org/10.7554/eLife.31134.008

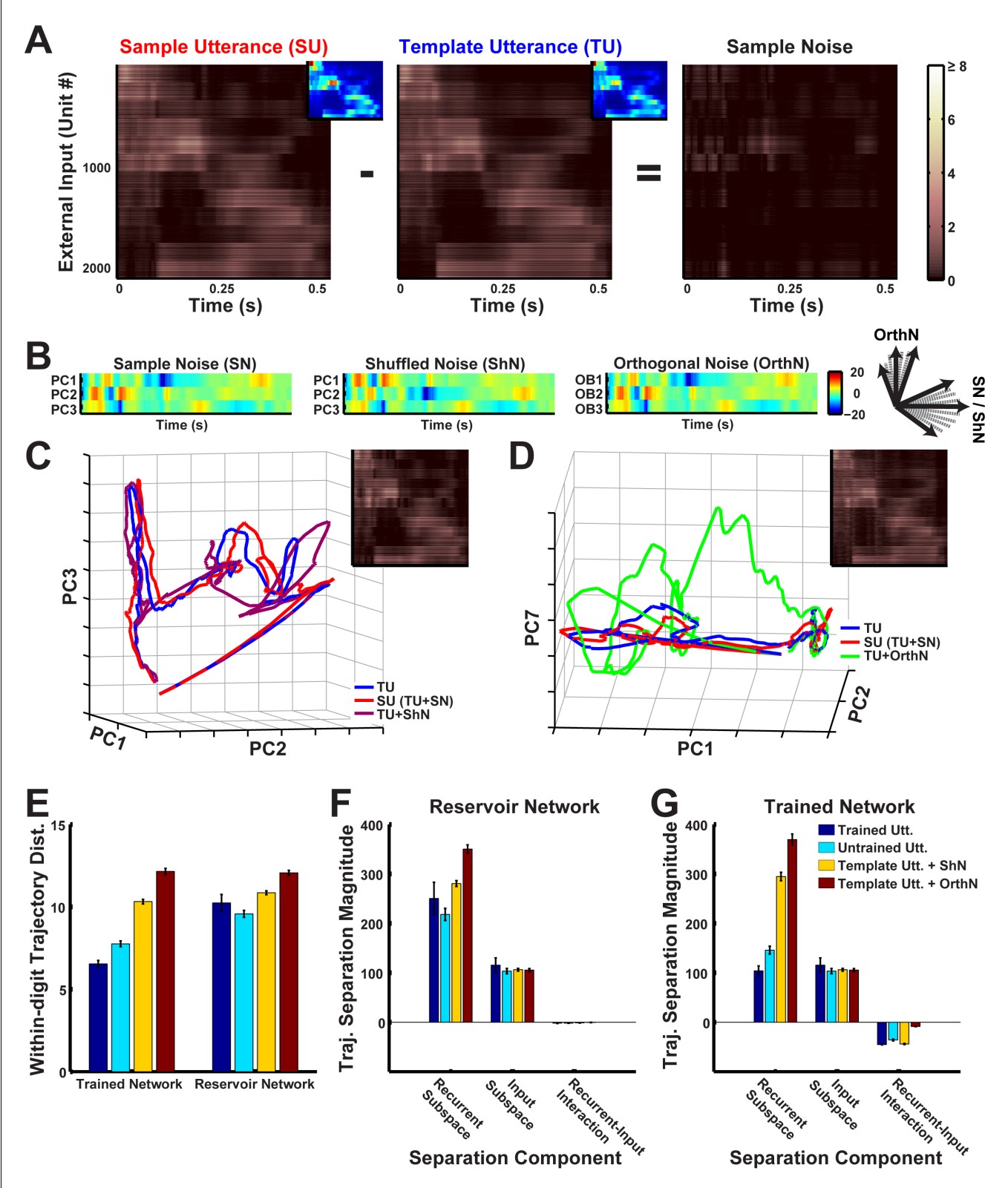

**Figure 6.** Robustness to spectral noise depends on the spatiotemporal structure of the inputs that the network is exposed to during training. (**A**) Spectral noise in the inputs to an RNN (N = 2100) during presentations of digit zero. Sample noise is the difference between the external input (each row reflects net external input to a unit in the RNN) for a sample and template utterance (corresponding cochleogram inputs are shown as insets; see Materials and Methods for the definition of template utterance). Absolute values of external inputs are shown here for better visualization. (**B**) *Figure 6 continued on next page*

*Figure 6 continued*

Coordinates of spectral noise in PCA space used to construct noisy utterances for digit zero. First three Principal component (PC) scores of the sample noise (left). Scores, where the basis was constructed as a random shuffle of the sample noise PC loadings (middle) and an orthonormal set orthogonal to the sample noise PC loadings (right). (C–D) Projections in PCA space of the external input for a template utterance, a sample utterance, and utterances with artificial noise created from the shuffled basis noise (C) and orthogonal basis noise (D). The respective artificial external inputs are shown as insets. (E) Comparison of mean within-digit distances of the trajectories for the different natural (two trained, blue and seven untrained, cyan) and artificial (30 shuffled, yellow and orthogonal, red each) utterances from the template utterance. Bars represent mean of the time-averaged distances for respective utterances of all digits (one subject, 10 digits). Error bars indicate standard errors of the mean over the digits. (F–G) Magnitudes of components that contribute to the total within-digit distance measured in (E) in the reservoir (F) and trained (G) networks. Bars represent mean of the time-averaged values of the components for respective utterances of all digits (one subject, 10 digits). Error bars indicate standard errors of the mean over the digits. The network was trained with $I_0$ set to 0.25, and trajectories were recorded with $I_0$ set to 0.

DOI: https://doi.org/10.7554/eLife.31134.009

The following figure supplement is available for figure 6:

**Figure supplement 1.** Schematic description of the decomposition of trajectories and trajectory separation into recurrent and input components.

DOI: https://doi.org/10.7554/eLife.31134.010

were generated by reordering the directions of spectral noise in phase space (*Figure 6C*). This procedure scrambled the relationships between input signal and noise, while constraining the artificial spectral noise to the same subspace as natural spectral noise. Orthogonal spectral noise was composed of linear bases that were orthogonal to natural spectral noise (*Figure 6D*, see Materials and Methods). In addition to scrambling the input signal-noise relationships, this procedure also scrambled the relationships between input channels. Care was taken while constructing these artificial inputs, to ensure that the temporal structure, dimensionality and magnitude of the spectral noise were identical to that observed in natural utterances.

Measurements of the within-digit trajectory distances revealed a progressive increase from trained utterances, to untrained utterances, to utterances with shuffled noise, to utterances with orthogonal noise (*Figure 6E*). In contrast, the untrained (reservoir) network exhibited deviations that were large and fairly similar across all conditions, implying that the trained network was able to identify and suppress natural spectral noise present in the utterances, but not artificial noise of similar magnitude and structure. We further dissected these results with a novel analysis based on the decomposition of the RNN drive into its input and recurrent components, which allowed trajectories to be decomposed into the subspaces wherein the network integrates external (input subspace) and recurrent (recurrent subspace) inputs (Materials and Methods). The analysis showed that in untrained networks, both natural and artificial spectral noise introduced deviations in the recurrent subspace that were large and unsuppressed (*Figure 6F*). In contrast, in trained RNNs the recurrent dynamics suppressed natural spectral noise, but not artificial noise (*Figure 6G*). Training also resulted in a recurrent subspace that suppressed spectral noise induced deviations in the input subspace. Whereas the untrained network integrated recurrent and external inputs in orthogonal subspaces, training the network rotated the recurrent subspace such that spectral noise induced trajectory deviations in the recurrent subspace were anti-correlated with the deviations in the input subspace, thereby resulting in their suppression (Recurrent-Input interaction in *Figure 6F–G*).

Together, these results suggest that recurrent plasticity may play a crucial role in identifying and suppressing the natural spatial variations of stimuli. In our model, this is achieved via supervised training with a common within-digit target trajectory, which exposes the spatiotemporal distribution of the spectral noise in the sensory input, and in doing so, allows for effective sampling from this distribution. Training then shapes the basins of attraction or hypertubes around the spoken digit-encoding trajectories based on this distribution, resulting in effective, recurrently-driven suppression of naturally occurring spectral noise.

## Encoding stimuli as neural trajectories allows for temporal generalization (Scaling)

As mentioned above, a signature and little understood feature of how the brain processes time-varying patterns pertains to temporal warping (temporal invariance), whereby temporally compressed or dilated input signals can be identified as the same pattern. Thus, an important test for computational models of sensory processing of time-varying stimuli, is whether they are able to account for

temporal invariance. We hypothesized that one advantage of encoding time-varying stimuli as continuous neural trajectories, is that it naturally addresses the problem of temporal warping; specifically, that compressed or dilated stimuli generate similar neural trajectories that play out at different 'speeds' (*Lerner et al., 2014*; *Mello et al., 2015*). To test this hypothesis, we trained and tested an RNN on a dataset of temporally warped spoken digits.

Specifically, the input pattern, $y(t)$, of a single utterance of each digit from a speaker of the TI-46 dataset was artificially stretched or compressed by a time warp factor $\alpha$ ($y(\frac{1}{\alpha}t)$), while retaining its spatial structure. *Figure 7A* (left panel) shows the input structure for two utterances of the digit 'nine', one warped to twice ($\alpha = 2$, 2x or 200% warp) and the other to half ($\alpha = 0.5$, 0.5x or 50% warp) the duration of the original utterance.

The RNN was trained on three warped utterances of each digit (warp factors of 0.7, 1 and 1.4) and tested on warp factors in the range 0.5 to 2 (a four-fold factor that approximates the natural speed variation of speech). To discern the effect of a temporally warped input on the encoding trajectory, we constructed a matrix of the distances between trajectories produced by the reference ($\alpha = 1$) and a warped utterance (*Figure 7A*). Each element of such a matrix measures the distance between the two trajectories at a corresponding pair of time points. In the case of perfect warping (e.g., when the reference trajectory and the trajectory at an $\alpha \neq 1$ overlap exactly in phase space), a diagonal line of zero distances (deep blue) would be observed during the sensory epoch (shaded region), with slope proportional to the warp factor. Trajectories produced by temporally warped inputs in the sensorimotor trained RNN were closer to the reference compared to their counterparts in the reservoir or motor-trained RNN. In other words, tuning the recurrent weights dramatically improved the ability of the network to reproduce the same neural trajectories, despite being driven by inputs at different speeds. We quantified this effect by measuring the average distance between the reference and warped trajectories during the sensory epoch over different warp factors (*Figure 7B*). In comparison to the two control networks, the trained RNN maintains a much smaller distance between trajectories across the four-fold range of temporal warping—including warp factors outside the range used during training. These results confirm the hypothesis that training produces a modulation of the internal dynamics by the external input that renders the encoding trajectories invariant to temporal warping of the input patterns. The impact of this result is borne out in the network's performance on the digit transcription task, as measured by the CNN classifier (*Figure 7C*). Training during the sensory and motor epochs dramatically improves 'extrapolation'— that is generalization to speeds outside the range of training speeds; however, training during the motor epoch alone is sufficient to produce very good 'interpolation' (novel speeds within the training set range) generalization.

## Mechanisms underlying temporal generalization (temporal scaling)

It is surprising that a RNN with strong recurrent dynamics can represent temporally scaled stimuli with similar trajectories. Specifically, the strong intrinsic dynamics of the network (governed by the recurrent weights) must somehow match the temporal scale of the input. One possibility is that the network achieves this by scaling the linear speed of its trajectories (i.e. the magnitude of $\frac{dx}{dt}$ in *Equation 1*) in inverse proportion to the warp factor. However, a comparison of the time-averaged linear speed of trajectories encoding digits at different warps deviates from this relationship in both the untrained and trained networks (*Figure 8A*), which, in contrast to our earlier results (*Figures 4A* and *7A*), suggests that the trajectories may not be temporally invariant. This contradiction arises from the incorrect assumption that the trajectories have little or no curvature. *Figure 8B* schematizes the two alternatives for temporal scaling of trajectories with curvature: constant speed/variable distance traversed, versus variable speed/constant distance traversed. At one extreme, a reference trajectory (black curve) can be slowed down while conserving its linear speed by increasing its radius of curvature, and therefore the distance it traverses through phase space (green curve). At the other extreme, the reference may be slowed by appropriately scaling down its linear speed while conserving its radius of curvature (yellow curve), thereby generating temporally invariant trajectories that traverse identical distances.

Temporally-scaled utterances produce overlapping trajectories in PCA subspace (*Figure 8C*, left panel)—reflecting the scaling of their linear velocities while traversing constant distances (a direct consequence of how the artificially warped utterances were constructed). In contrast, the RNN

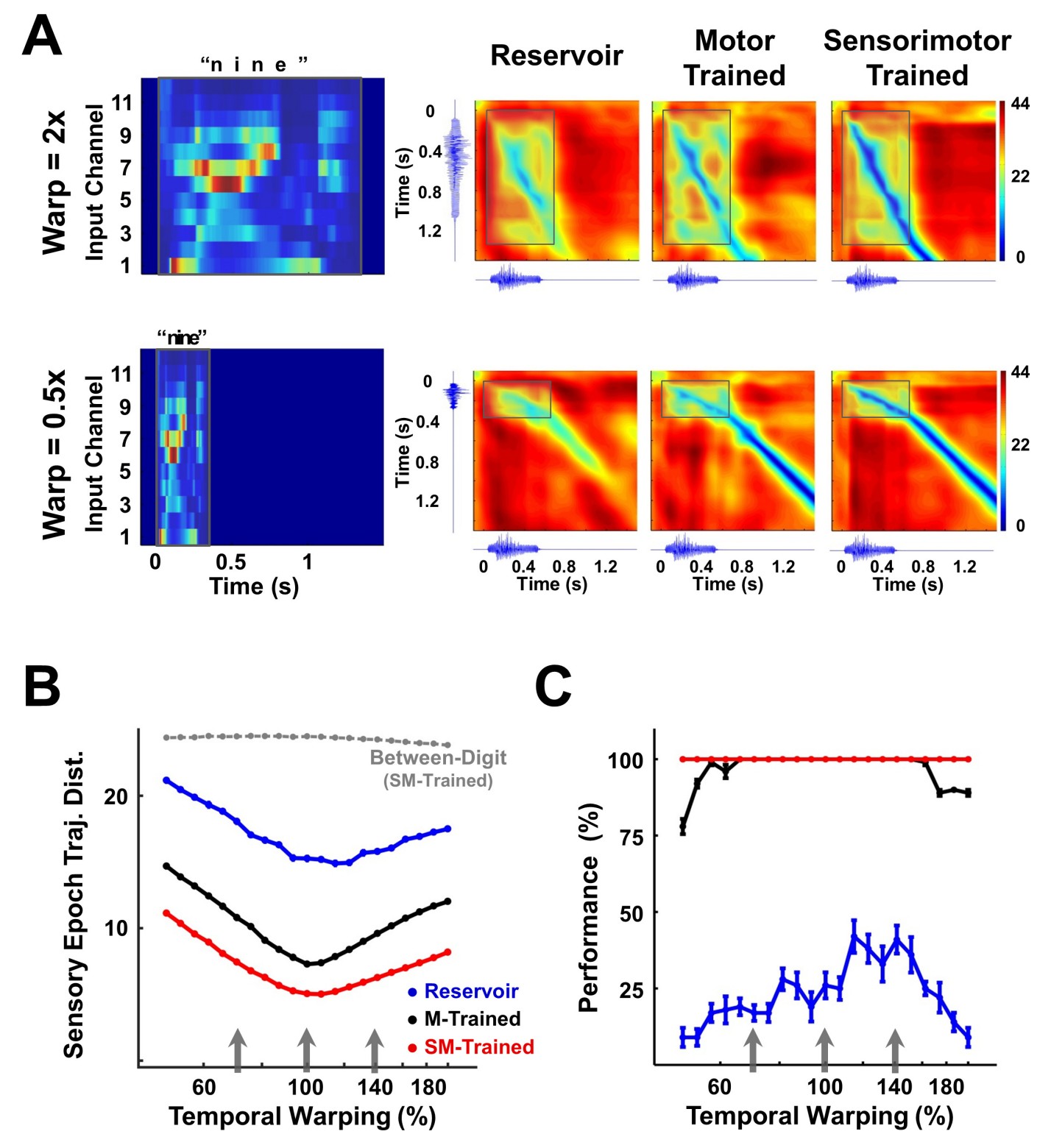

**Figure 7.** Invariance of encoding trajectories to temporally warped spoken digits. (**A**) Temporally warped input cochleograms for an utterance of the digit 'nine' (left panel), warped by a factor of 2x (upper row) and 0.5x (lower row). Distance matrices between the trajectories produced by the warped and reference stimuli for the two control networks and the sensorimotor trained network (right panels). Distance matrices are accompanied by the corresponding sonograms on each axis, and highlight the comparison between the sensory trajectories of the warped and reference utterance (outlined box). A deep blue (zero-valued) diagonal through a distance matrix indicates that the sensory trajectory encoding the warped input overlaps with the

*Figure 7 continued on next page*

*Figure 7 continued*

reference trajectory. Thus, the sensorimotor trained RNN is more invariant (deeper blue diagonal) to temporal warping of the input than the motor-trained and reservoir RNNs. (**B**) Mean time-averaged Euclidean distance between sensory trajectories encoding warped and reference utterances of the 10 digits in each of the three networks, over a range of warp factors. Dashed gray line indicates mean distance between sensory trajectories encoding warped and reference utterances of different digits in the sensorimotor trained network. Error bars (not visible) indicate standard errors of the mean over 10 test trials. Gray arrows indicate the warp factors at which the networks were trained. (**C**) Transcription performance (measured by the deep CNN classifier) of the three networks on utterances warped over a range of warp factors. The results corroborate the measurements in (**B**) and show that a sensorimotor trained RNN is more resistant to temporal warping. All networks were composed of 2100 units. Network training was performed with 30 utterances (one subject, 10 digits, one utterance per digit, three warps per utterance). $I_0$ was set to 0.25 during network training, and to 0.05 during output training and testing. Reference trajectories for the distance analysis were recorded with $I_0$ set to 0.

DOI: https://doi.org/10.7554/eLife.31134.011

trajectories produced by these scaled utterances are spatially distinct with warp-dependent separations (*Figure 8C*, right panel). Moreover, they exhibit warp-dependent curvature, with slower trajectories incurring larger curvature radii. Together with the results in *Figure 8A*, this suggests that the mechanisms underlying temporal scaling in our model fall between the two extreme hypotheses, implying that: (i) that the average linear speed and curvature radius of trajectories encoding utterances at different warps adjust in a manner that appropriately modulates their angular speed and therefore their timescale; and (ii) that the trajectories are parallel to each other, demonstrating that they encode the digit similarly. *Figure 8D* presents measurements of the time-averaged linear speed and the total distance traversed in the recurrent subspace of phase space, by the trajectories in the trained and untrained networks over a range of warps. The latter is a heuristic measure of the time-averaged curvature radius, hinging on the fact that the circumference (or distance traversed) linearly scales with the radius. A comparison of these measurements for the two networks to the expected values in constant speed- and constant distance trajectories, confirms that both networks produce trajectories that lie in between these two extremes. That is, temporal scaling is achieved by an approximate balance of both hypotheses, e.g., at low speeds (digits spoken slowly) the trajectories are slower and longer. Furthermore, training alters this balance to reduce the within-digit separation, by modulating the trajectory speeds more strongly. Measurements to establish whether the trajectories are parallel also agree with these results (*Figure 8—figure supplement 1C*).

But how do identical inputs (except for their duration) generate these parallel trajectories (e.g., *Figure 8C*)? To answer this question, we first measured the relationship between the input and recurrent subspaces in the trained network, and discovered that they are orthogonal to one another (data not shown), implying that the integration of the external inputs is independent of the integration of the recurrent inputs. In the absence of subspace interactions, it is easy to see that external inputs with different speeds are integrated into spatially distinct input subspace trajectories (e.g. $\int \cos(\omega t)dt = \frac{1}{\omega}\sin(\omega t) + c$, wherein high speed/frequency signals integrate to produce trajectories with small magnitudes) (*Figure 8—figure supplement 1A*). As with the recurrent subspace trajectories, the resulting input subspace trajectories are parallel to each other and modulate their angular speed via a combination of changes to their curvature radii and linear speed (*Figure 8—figure supplement 1A–C*). Ultimately, the differentiated curvature and linear velocity that modulate the angular velocity and therefore the timescales of recurrent subspace trajectories, are directly shaped by these input subspace dynamics (*Rajan et al., 2010a*).

To determine if this solution to the temporal invariance problem is specific to the 'innate' training algorithm, or likely a general property of trained RNNs, we trained a RNN with a gradient-descent approach in which the error was based solely on the output units (Materials and Methods, *Figure 8—figure supplement 2*). While the interpolation and extrapolation performance of this network was inferior to the network trained with 'innate' learning (*Figure 8—figure supplement 2B*), the underlying mechanism was the same: warped utterances were encoded along parallel trajectories (*Figure 8—figure supplement 2F–G*) with warp-dependent curvature and linear velocity (*Figure 8—figure supplement 2E*) that subserved temporal scaling. Thus, two very different training algorithms resulted in similar solutions to the temporal invariance problem.

In summary, these results demonstrate that the integration of inputs at different warps separates out the resulting trajectories in the input subspace, and consequently in the recurrent subspace. In both subspaces the network dynamics generate trajectories with warp-dependent linear velocities

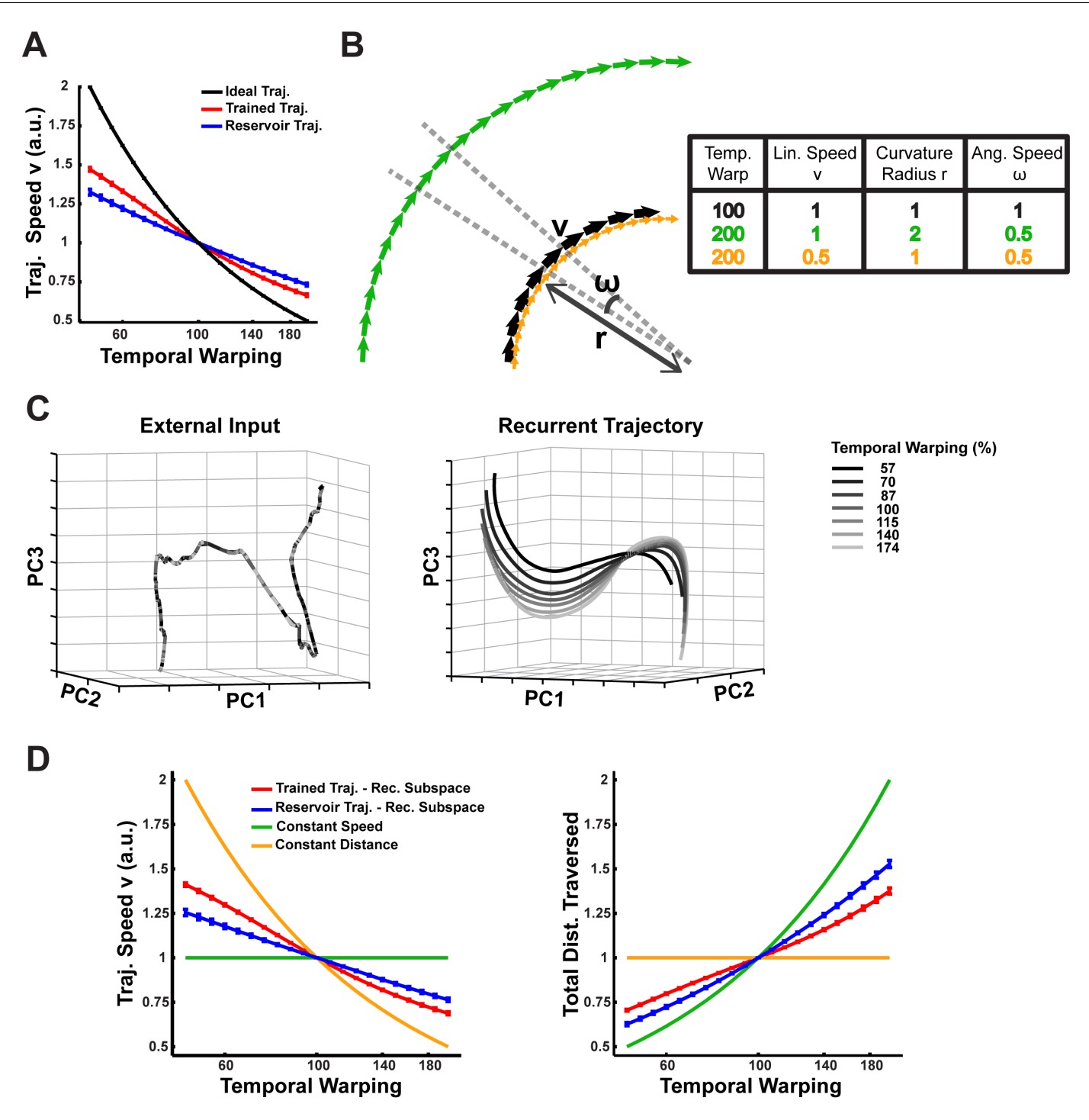

**Figure 8.** Mechanism of temporal scaling invariance. (**A**) Time-averaged linear speed (**v**) during the sensory epoch trajectories in the reservoir and trained networks (N = 2100) compared to the ideal linear speed, over a range of warp factors. The speeds are normalized to the 1x warp. (**B**) Schematic representation of two hypotheses for trajectories that exhibit temporal warp invariance via warp-dependent angular speeds (ω): constant linear speed/variable distance (green) and variable linear speed/constant distance (yellow) (**C**) Projections in PCA space of external input for warped utterances of digit one (left) and the corresponding sensory epoch trajectories of the trained network (right), over a 100 ms duration of the 1x warp trajectory and the corresponding utterance interval of the remaining trajectories. Over this short duration, over 90% of the variance in the external input and the trajectories was captured. (**D**) Mean linear speed (left) and cumulative distance traversed (time integral of the linear speed; right) in the recurrent subspace of phase space of a reservoir and trained network, compared to predictions from the constant speed and constant distance hypotheses. The

*Figure 8 continued on next page*

*Figure 8 continued*

measures are normalized to the 1x warp. In all panels error bars indicate SEM over the digits. The network was trained with $I_0$ set to 0.25, and trajectories were recorded with $I_0$ set to 0.

DOI: https://doi.org/10.7554/eLife.31134.012

The following figure supplements are available for figure 8:

**Figure supplement 1.** Phase space relationships in the input and recurrent subspaces as a function of the warp factor.

DOI: https://doi.org/10.7554/eLife.31134.013

**Figure supplement 2.** Temporal invariance performance and mechanism in a network trained with Backpropagation through time (BPTT).

DOI: https://doi.org/10.7554/eLife.31134.014

and curvature, which appropriately modulate their angular velocity, resulting in them traversing spatially distinct yet parallel paths at warp-dependent timescales. This confirms the hypothesis that an inherent computational advantage of encoding time-varying sensory stimuli in neural trajectories is that it can naturally address the temporal invariance problem.

## Discussion

The brain naturally encodes, recognizes, and generates complex time-varying patterns, and can seamlessly process temporally scaled patterns. Despite our poor understanding of these processes, theoretical evidence increasingly suggests that the dynamics of recurrently connected circuits in the brain are critical to representing time-varying patterns. For example, so-called reservoir computing approaches propose that complex high-dimensional spatiotemporal patterns are represented in the dynamics inherent to randomly connected recurrent neural networks (*Jaeger and Haas, 2004*; *Buonomano and Maass, 2009*; *Sussillo and Abbott, 2009*). A shortcoming of this approach, however, is that the recurrent connections in these networks are not plastic; thus in contrast to actual cortical circuits, the random recurrent neural network (the 'reservoir') does not adapt or optimize to the task at hand. Additionally, fully tapping into the computational potential of randomly connected RNNs has proven difficult because they are susceptible to chaos (*Sompolinsky et al., 1988*; *Wallace et al., 2013*). However, progress has been made on both accounts (*Martens and Sutskever, 2011*; *Vogels et al., 2011*; *Laje and Buonomano, 2013*). Here, we extend these results to better understand well-known aspects of the brain's ability to represent time-varying sensory *and* motor patterns—we demonstrate how experience-dependent plasticity could reshape the dynamics of reservoir RNNs to qualitatively improve representations by endowing them with such essential properties as generalization to novel exemplars, sensory-motor association and transformation, and temporal invariance.

Previous models of temporal scaling have enlisted neural and synaptic dynamics to achieve temporal invariance of both sensory stimuli (*Gütig and Sompolinsky, 2009*) and the generation of motor responses (*Murray and Escola, 2017*). In the model proposed by Gutig and Sompolinsky, temporally invariant recognition of compressed and dilated sensory stimuli is based on a synaptic shunting mechanism, wherein the effective integration time of a post-synaptic cell is modulated by the total conductance of its afferent synapses, which changes as a monotonic function of the input warp—thus temporal invariance is ultimately reduced to a cellular property. In contrast, in the model described here temporal invariance emerges from a network property: the formation of parallel neural trajectories traversed at different speeds. While the formation of such trajectories may be surprising given the sensitivity and nonlinear nature of RNNs, we show that this capacity is the consequence of scale-dependent angular velocities of the encoding trajectories. This property of RNN encoding, which is observable to a lesser extent even in reservoir networks (*Figure 8D*), stems from its leaky integration of scaled inputs (*Figure 8—figure supplement 1B*) and the input-driven suppression of its chaotic recurrent activity (*Rajan et al., 2010a*). Indeed, network training enhances this property by suppressing recurrent amplification of scale-driven deviations in the encoding trajectories, producing proximal, parallel within-digit trajectories with linear velocities that are more strongly scale-dependent (*Figure 8D*, *Figure 8—figure supplements 1C*, *2E and G*). The effects of such a suppression are evident even in networks subjected to motor-only training, where we observe a considerable decrease in the within-digit distances (*Figure 7B*) and consequently strong interpolation generalization (*Figure 7C*). Nevertheless, it remains the case that training across both the

sensory and motor epochs, not only improves interpolation and extrapolation generalization, but also strongly enhances cross-utterance and cross-speaker generalization (*Figure 1C*).

Our results also demonstrate how training an RNN reveals a computationally powerful pattern recognition regime, one that helps generalize the encoding of learned time-varying patterns to novel exemplars. As stated earlier, recognizing different instances of the same digit as one and the same, while discriminating between different digits requires that networks actively and differentially sculpt the within and between class trajectories—the separation between trajectories representing similar patterns must be kept to a minimum, while that between trajectories representing different patterns must be amplified. A previous study has analytically shown that untrained random recurrent networks (reservoir networks) process inputs in one of three computational regimes: a chaotic regime, wherein any separation between trajectories is strongly amplified by the network's chaotic dynamics; a regime with weak recurrent weights that is input-dominated with little computational power; and a critical regime wherein the input and internal dynamics are balanced (*Bertschinger and Natschläger, 2004*). Here, we show that training alters these regimes in very different ways (*Figure 5*). Crucially, relative to the separation between the input patterns, reservoir networks in the critical regime amplify the separation between trajectories representing similar patterns more strongly than between different patterns. In contrast, trained networks separate trajectories according to the patterns they are representing (*Figure 5A*). Tuning the recurrent weights reshapes the internal dynamics of the network, enabling it to redirect or suppress deviations from the target trajectories (*Figure 6*). Ultimately, it is this property of trained networks that allows them to encode complex sensory patterns and effectively generalize across natural utterances and speakers.

Three experimentally tractable predictions emerge from the above results. First, neural trajectories will be stable in response to local perturbations potentially administered through optogenetic stimulation. Importantly, and in agreement with an earlier analysis (*Rajan et al., 2010a*), the neural trajectories elicited during sensory stimuli will be more resistant to perturbations than the neural trajectories unfolding during the motor epoch of sensory-motor tasks. A second prediction, which to the best of our knowledge is a novel one, relates to the structure of noise. Specifically, trained networks fail at generalizing to unfamiliar spectral noise patterns, thereby generating the prediction that trajectories in sensory areas will diverge more when presented stimuli composed of artificial or unfamiliar spectral noise patterns. This would indicate that the stimulus-response function of complex sensory neurons are not just sensitive to natural stimulus features (*Theunissen and Elie, 2014*), but also natural noise features. To test this, an animal could be trained to recognize natural sounds with noise injected along some principal components of the stimulus spectrogram but not others, and then tested on noise injected along the held-out PCA dimensions. A comparison of the trajectories encoding trained and tested sounds should reveal a larger divergence in the neural encoding of the test sounds. Third, and most specifically, the model predicts a clear neural correlate of the encoding of temporally scaled stimuli—the observation that slower stimuli yield trajectories with larger curvature radii implies that the neural population activity should exhibit larger fluctuations in their firing rates in response to slower stimuli; In other words, at the neuronal level the range of minimal to maximal firing rate should be larger (*Figure 8—figure supplements 1D* and *2D*).

These experimental predictions are critical to validate the model's implementation of complex and invariant sensorimotor computations as stable neural trajectories. However, even if validated a number questions remain to be addressed. Most notably, how can the recurrent synaptic strengths be tuned to develop stable trajectories in a biologically plausible manner? The learning rule used here, coupled with its requirement of an explicit target for each recurrent unit, make it biologically implausible. However, while it is important that the sensorimotor representations are encoded as locally stable trajectories, the structure of the target trajectories themselves are essentially arbitrary. It is therefore conceivable that arbitrary yet locally-stable encoding trajectories may emerge from unsupervised learning. A second related issue is that to achieve strong temporal invariance, our model had to be trained over a range of sensory stimuli speeds. Again, it is not clear if this would represent a biologically plausible scenario—to help address this question, it will be important for future research to determine if the ability to recognize stimuli independent of speed is learned through experience. Finally, questions relating to the learning capacity of networks capable of strong temporal invariance, and the expedience of sensory-epoch training for temporal invariance remain open.

The computational potential of continuous time RNNs in high-gain regimes has long been recognized, but it has been challenging to tap into this potential because of their inherently chaotic behavior and the difficulties in training them (*Bengio et al., 1994*; *Pearlmutter, 1995*). Step-by-step, progress has been made on how to capture the computational potential of RNNs (*Jaeger and Haas, 2004*; *Sussillo and Abbott, 2009*; *Martens and Sutskever, 2011*; *Laje and Buonomano, 2013*). Here, we establish that a further feature of trained RNNs is their ability to not only encode spatiotemporal objects, but also to perform complex sensorimotor tasks and address the long-standing problem of temporal warping. These results are consistent with the proposal that that spatiotemporal objects are not encoded as fixed-point attractors, but as locally stable neural trajectories (dynamic attractors).

## Materials and methods

### Network model

The dynamics of the RNN was comprised of $N$ nonlinear continuous-time firing rate units modeled as:

$$\tau\frac{dx_i}{dt} = -x_i + \sum_{j=1}^{N} W_{ij}^R r_j + \sum_{j=1}^{M} W_{ij}^I y_j + I_i^{noise} \tag{1}$$

$$r_i = tanh(x_i) \tag{2}$$

$x_i$ represents the state of neuron $i$, and $r_i$ its 'firing rate'. The time constant, $\tau$, of each unit was set to 25 ms. $\boldsymbol{W^R}$ is the weight matrix representing the recurrent connectivity of the network. The recurrent connectivity was uniformly random (but with no autapses) with a connection probability ($p_c$) of 0.2. The weights of these synapses were initialized from an independent Gaussian distribution with zero mean and SD equal to $g/\sqrt{p_c N}$, where $g$ represents the 'gain' of the network. RNNs were initialized to a high-gain regime ($g$ = 1.6), which generates chaotic self-perpetuating activity in the absence of external input or noise (*Sompolinsky et al., 1988*).

The M-dimensional vector $\boldsymbol{y(t)}$ represents the time-varying sensory input to the RNN. The fixed input weight matrix, $\boldsymbol{W^I}$, tonotopically projects this input vector onto the RNN—input channel $k$ is projected onto units (k-1)×N/M + 1 thru k × N/M. The weights of these connections were drawn from an independent Gaussian distribution with zero mean and unit variance. $\boldsymbol{W^I}$ $\boldsymbol{y(t)}$ represents the net external input to each RNN. The handwriting was modeled with three output units ($o_1$, $o_2$, $o_3$) that represented the x, y and z co-ordinates of a pen on paper. Finally, each unit of the RNN also received an independent background noise current, $\boldsymbol{I^{noise}}$, modeled as additive Gaussian white noise with SD $I_0$.

As is standard, the output neurons were simulated as a weighted linear sum of all the units in the RNN:

$$o_i = \sum_{j=1}^{N} W_{ij}^O r_j \tag{3}$$

where the output weight matrix, $\boldsymbol{W^O}$, was initialized from an independent Gaussian distribution with zero mean and SD $1/\sqrt{N}$. All simulations were performed with a time step of 1 ms.

### Simulations and training

Each trial consisted of a time window comprised of a sensory and a motor epoch. We defined the sensory epoch as the period in which an external stimulus is presented—starting at $t$ = 0 and lasting the duration of the utterance. The motor epoch was defined as a period beginning 300 ms after the sensory epoch ended and lasting the duration of the target motor pattern (the appropriate handwritten digit).

Training proceeded in three steps (described in detail below): (1) target trajectories were generated for each utterance of each digit in the training subset of the dataset; (2) the recurrent units were trained to reproduce the target trajectories; and (3) the output units were trained to produce

the handwritten spatiotemporal patterns. The trained network was then tested on novel (and trained) utterances. In steps 2, 3 and during testing, each trial began at $t$=-100 ms with the network initialized to a random state ($x_i$ values drawn from a uniform distribution between $-1$ and 1). In *Figures 1* and *7*, *Figure 1—figure supplement 1* and *Figure 8—figure supplement 2A–2B*, testing was performed with the noise amplitude ($I_0$) set to 0.05. For the perturbation analysis (*Figure 2*), a 25 ms 'perturbation pulse' was introduced during the sensory or motor epoch of each trial, with $I_0$ set to the desired perturbation magnitude for the duration of the pulse. Noise was omitted in these simulations to allow for a direct assessment of the effects of the perturbation pulse. Similarly, for the simulations in *Figures 3–6*, *Figure 8* and *Figure 8—figure supplements 1* and *2C–2G*, noise was omitted ($I_0 = 0$) so that the impact of training on the generalization and discriminability of an RNN's encodings, but not its background noise invariance, could be direct evaluated.

Simulations of the untrained 'reservoir' control network skipped steps 1 and 2, while simulations of the motor-trained control network limited recurrent network training (step 2) to a duration starting 150 ms after the end of the sensory epoch and lasting until the end of the motor epoch.

## Input structure

We used spoken digits from the TI-46 spoken word corpus (*Mark Liberman et al., 1993*) to train and test networks on the transcription task. Specifically, our dataset was composed of the spoken digits 'zero' thru 'nine', uttered 10 times each, by each of five female subjects. Spoken digits from the corpus were decoded, end-pointed, resampled to 12 kHz, and converted to spectrograms with Matlab's specgram function. The spectrograms were preprocessed with Lyon's passive ear model of the human cochlea (*Lyon, 1982*), as implemented by the auditory toolbox (Malcolm Slaney), to generate analog 'cochleograms' composed of 12 analog frequency bands, or channels, ranging from 0 to 6 kHz. Finally, the cochleograms were smoothed with a $20^{th}$ order 1D median filter (Matlab's medfilt1 function), normalized to a maximum of 1 (i.e., they were normalized by the maximal value of all utterances and digits), and scaled by an input amplitude. During the sensory epoch of each trial, the input, $y(t)$ (M = 12), took values from the cochleogram corresponding to the utterance that was to be presented to the RNN. At all other times of the trial, $y(t)$ was set to 0. The input amplitude was set to five in all simulations except in *Figure 5*, where it was parametrically varied. For the temporal warping simulations (*Figures 7–8*, *Figure 8—figure supplements 1–2*) the cochleograms were compressed or dilated by the warping factor $\alpha$ through linear interpolation.

## Innate trajectories and RNN training

Training was performed with the 'innate-learning' approach—which uses the Recursive Least Squares (RLS) rule to train each unit of the recurrent network to match the pattern generated by the untrained network (*Laje and Buonomano, 2013*). For each subject used in the training set, a single template utterance of each digit was presented to the untrained RNN (the reservoir) in the absence of background noise, and the resulting trajectory served as a target ('innate') trajectory. The template utterance of each digit was chosen as one with the median duration among all the utterances of the digit by the subject. Target sensory epoch trajectories for other training utterances of the digit by the same subject were generated by linearly warping this innate trajectory to match the utterance durations.

To achieve the formation of digit-specific dynamic attractors, a single target motor trajectory was adopted across subjects and utterances for each digit. This target, comprising of the sensory-to-motor transition and motor epoch population activity, was generated from the template utterance by a single arbitrarily chosen subject, by allowing the untrained RNN to autonomously generate activity following template stimulus offset, in the absence of background noise, for a duration equal to the sum of the sensory-to-motor transition duration (300 ms) and the motor epoch duration for the digit. Sensorimotor targets for each training utterance were composed by concatenating the common motor target to the end of the corresponding sensory targets. The initial network state was chosen at random while harvesting the target trajectory for each digit. Target trajectories for template utterances of the same digit by different speakers were harvested starting the network at the same initial state.

All networks were trained with three utterances of each digit for each of the subjects in the training set. The networks in *Figure 1* and *Figure 1—figure supplement 1* were trained on three

subjects and tested on five, while those in *Figures 2–8* and *Figure 8—figure supplements 1–2* were trained and tested on one subject. The network size, $N$, strongly impacted its capacity, particularly for generalization. Accordingly, we trained larger networks to robustly address cross-speaker generalization ($N = 4000$) relative to those trained for cross-utterance generalization within a single speaker ($N = 2100$).

Network training was performed by modifying the recurrent weight matrix, $W^R$, with the Recursive Least Squares learning rule (*Haykin, 2002*). The rule was simultaneously applied to 90% of the units in the network (randomly selected). Training was conducted by iterating through all utterances of the digits in the training set over multiple trials, starting each trial at a random initial condition and continuously injecting the network with background noise ($I^{noise}$). Training concluded when the error in the activity of the rate units asymptoted (generally between 100 and 150 trials).

## Output training

The spatiotemporal target patterns that comprise the handwritten digits were sampled from a Wacom Bamboo Pen Tablet, as the digits '0' thru '9' were individually stenciled on it. For each handwritten digit, the x and y coordinates of the pen were sampled at approximately 50 Hz, low-pass filtered, and resampled with interpolation to 1 kHz (corresponding to the 1 ms simulation time step). The target values for $o_1$ and $o_2$ were set to 0 from the start of each trial until the beginning of the motor epoch. During the motor epoch, they were set to the pen's 2D coordinates for the corresponding digit, and reset to 0 between the end of the motor epoch and the end of the trial. The target for $o_3$ (z co-ordinate) was a step function, set to one during the motor epoch and 0 at all other times. To train the output units, the Recursive Least Squares learning rule was applied to the readout weights in $W^O$ (*Haykin, 2002*; *Jaeger and Haas, 2004*; *Sussillo and Abbott, 2009*; *Laje and Buonomano, 2013*). Output training was performed for 25 trials per utterance of each digit in the training set, while the RNN was continuously injected with background noise.

In each test trial, the motor output was recorded from the values of $o_1$ and $o_2$, whenever the value of $o_3$ was greater than 0.5 (i.e. when 'the pen contacted the paper'). At the end of the trial, a $28 \times 28$ pixel grayscale image of the 'handwritten' output (pen width = 2 pixels) was labeled as the transcription for the corresponding digit. An objective determination of the transcription's accuracy was made by comparing this label to one assigned to the image by a CNN classifier for handwritten digits. This classification was performed with the LeNet-5 CNN classifier (*Lecun et al., 1998*) implemented with the Caffe deep learning framework (*Jia et al., 2014*). The CNN was first trained on the MNIST database of handwritten digits (*LeCun et al., 1998*), and its output layer was then fine-tuned on the stenciled digits that we used as the output targets.

## Gradient descent training

In *Figure 8—figure supplement 2*, we evaluate the mechanism underlying temporal scaling in a network trained with gradient descent. The recurrent rate units and output units were modeled as in *Equations 1-3*, with weights of the input and output units initialized as in the reservoir network. Target values for the output units were assigned as described above. No target values were specified for the recurrent rate unit activity. Instead, all input, recurrent and output weights of the network were updated at the end of each training trial, with weight update values for the trial determined based on the squared-error of the outputs averaged across time and output units. Specifically, the error was transformed into weight updates via the backpropagation through time (BPTT) algorithm applied via ADAM optimization (*Kingma and Ba, 2014*). To evade the vanishing and exploding gradients problem that commonly hinder gradient descent training of RNNs, $W^R$ was initialized as an all-to-all orthogonal matrix of random values (*Le et al., 2015*), and gradient magnitudes were restricted via gradient clipping (*Graves, 2013*). Furthermore, $W^R$ was initialized to high-gain regime ($g = 1.6$), as this lead to faster convergence (5000 trials per utterance). The network was trained with continuous injection of background noise. Training was performed with the Tensorflow library (*Abadi et al., 2016*), and output performance was evaluated with the CNN classifier described above.

## Trajectory analysis

In *Figures 4–5*, the sensory (motor) epoch within-digit distances were calculated from the Euclidean distance, at each time step, between the sensory (motor) epoch trajectory for each utterance, and its nearest neighbor from among the trajectories produced by the training utterances of the same digit. Similarly, the sensory (motor) epoch between-digit distances were calculated from the Euclidean distance, at each time step, between the sensory (motor) epoch trajectory for each utterance, and its nearest neighbor from among the sensory (motor) epoch trajectories encoding training utterances of all other digits. Similarly, in *Figure 6E*, sensory epoch within-digit distances were calculated as the Euclidean distance between the sensory epoch trajectory for each tested external input, and the one encoding the template utterance. Finally, trajectory distances in *Figure 7B* and *Figure 8—figure supplement 2A* were also calculated in a similar fashion: at each time step, the Euclidean distance was calculated between the trajectory encoding an utterance at warp factor $\alpha$ ($l_0 = 0.05$), and its nearest neighbor along the trajectory encoding the reference utterance of the same digit ($l_0 = 0$). The time-average of these distances were then summarized over 10 trials for the plots in these figures.

The nature and robustness of spectral generalization in trained RNNs was probed in *Figure 6*, by artificially altering the external inputs. In order to assess spectral generalization independent of temporal invariance, for each digit, the duration of all its utterances were time-normalized by warping the respective cochleograms to their median duration. The RNN and its outputs were trained on three time-normalized utterances for each of the ten digits, and all test simulations were performed with a common initial state. The spectral generalization of an RNN was tested with artificially constructed external inputs to the network that were qualitatively similar to external inputs for time-normalized natural utterances, but with altered spectral noise structure. The spectral noise in an utterance was defined as the difference between the external inputs of the utterance and the respective template utterance (*Figure 6A*). Network responses to time-normalized trained and untrained natural utterances were compared to two artificial controls: external inputs with shuffled noise and with orthogonal noise. Inputs with shuffled noise were constructed from the Principal Component Analysis (PCA) loadings and scores of the spectral noise in untrained natural utterances as follows: (i) the loading vectors were permuted; (ii) shuffled noise was generated by multiplying the spectral noise scores by the shuffled loading vectors; (iii) this shuffled noise was added to the external input of the corresponding template utterance. Inputs with orthogonal noise were also constructed from the PCA loadings and scores of the spectral noise in untrained natural utterances, except instead of permuting the loading vectors, they were replaced by an orthonormal set of vectors, generated via Gram-Schmidt orthogonalization, each of which was orthogonal to the set of spectral noise PCA loading vectors. For this a random vector was generated and QR factorization was performed on the set of vectors comprised of the PCA loadings and the random vector, producing a new vector that was orthogonal to the PCA loadings. This procedure was repeated to generate a set of such vectors equal in cardinality to the PCA loading set. The random vectors were then orthogonalized relative to each other in a similar manner, and then normalized to produce an orthonormal set. External inputs with shuffled and orthogonal noise were constructed from each untrained natural utterance, with as few PCs as were necessary to explain 99% of the variance in its spectral noise (typically 11 PCs). For each digit, the network was tested with 30 shuffled and orthogonal noise inputs, each based on a randomly chosen untrained natural utterance.

The dimensionality measure shown in *Figure 5* was calculated as $\left(\sum_{k=1}^{N} \lambda_k^2\right)^{-1}$, where $\lambda_k$ represents eigenvalues of the equal-time cross-correlation matrix of network activity, expressed as a fraction of their sum (*Rajan et al., 2010b*). The eigenvalues were calculated on a concatenation of the sensory epoch trajectories for all utterances (trained and novel) of all digits.

## RNN decomposition

In *Figure 6*, *Figure 8* and *Figure 8—figure supplements 1–2*, trajectories and their distances from each other were decomposed into the constituent input and recurrent subspaces of phase space. These are derived from the state variable *x(t)* (*Equation 1*), rather than the rate variable *r(t)* (*Equation 2*) that was used to measure Euclidean distances:

From Euler's Method:

$$x(t) = x(t-1) + \frac{dx(t-1)}{dt}$$

$$x(t) = \frac{\tau-1}{\tau}x(t-1) + \frac{1}{\tau}W^R r(t-1) + W^I y(t-1)$$

Let $a = \frac{\tau-1}{\tau}$, $b = \frac{1}{\tau}$, $R(t) = W^R r(t-1)$, and $I(t) = W^I y(t-1)$

$$x(t) = ax(t-1) + bR(t) + bI(t)$$

Solving this recurrence relationship,

$$x(t) = a^t x(0) + b\sum_{k=1}^{t} a^{t-k} R(k) + b\sum_{k=1}^{t} a^{t-k} I(k) \tag{4}$$

where $x(0)$ is the initial state of the network. We denote the first two terms $(a^t x(0) + b\sum_{k=1}^{t} a^{t-k} R(k))$ as network activity in the recurrent subspace (recurrent subspace trajectory) and the last term $(b\sum_{k=1}^{t} a^{t-k} I(k))$ as network activity in the input subspace (input subspace trajectory).

When the input durations for all utterances of a digit are time-normalized, as in *Figure 6*, the squared Euclidean distance between the state variables $x^1(t)$ and $x^2(t)$ for two sensory epoch trajectories encoding utterances $y^1(t)$ and $y^2(t)$ is given by:

$$\Delta x(t).\Delta x(t) = \left(x^2(t) - x^1(t)\right).\left(x^2(t) - x^1(t)\right)$$

From *Equation 4*, it follows that:

$$\Delta x(t) = a^t x^2(0) + b\sum_{k=1}^{t} a^{t-k} R^2(k) + b\sum_{k=1}^{t} a^{t-k} I^2(k) - a^t x^1(0) - b\sum_{k=1}^{t} a^{t-k} R^1(k) - b\sum_{k=1}^{t} a^{t-k} I^1(k)$$

Assuming common initial state and rearranging:

$$\Delta x(t) = b\sum_{k=1}^{t} a^{t-k}\left(R^2(k) - R^1(k)\right) + b\sum_{k=1}^{t} a^{t-k}\left(I^2(k) - I^1(k)\right)$$

Let $\overline{\Delta R(t)} = b\sum_{k=1}^{t} a^{t-k}\left(R^2(k) - R^1(k)\right)$ and $\overline{\Delta I(t)} = b\sum_{k=1}^{t} a^{t-k}\left(I^2(k) - I^1(k)\right)$ (*Figure 6—figure supplement 1*). Then,

$$\Delta x(t).\Delta x(t) = \left(\overline{\Delta R(t)} + \overline{\Delta I(t)}\right).\left(\overline{\Delta R(t)} + \overline{\Delta I(t)}\right)$$

$$\Delta x(t).\Delta x(t) = \overline{\Delta R(t)}.\overline{\Delta R(t)} + \overline{\Delta I(t)}.\overline{\Delta I(t)} + 2\overline{\Delta R(t)}.\overline{\Delta I(t)} \tag{5}$$

where the first $(\overline{\Delta R(t)}.\overline{\Delta R(t)})$ and second $(\overline{\Delta I(t)}.\overline{\Delta I(t)})$ terms denote the squared distance in recurrent subspace and input subspace, respectively. The third term $(2\overline{\Delta R(t)}.\overline{\Delta I(t)})$ is based on the interaction between the two subspaces: If the recurrent and input subspaces are orthogonal to each other, then this term will be zero; otherwise, it indicates whether the deviations in the recurrent subspace serve to amplify or suppress the spectral noise.

In *Figure 8* and *Figure 8—figure supplements 1–2*, all test simulations were performed with a common initial state. The linear speed of the trajectory in neural phase space, and in its input and recurrent subspaces, were calculated as the time-averaged magnitude, or $L_2$-norm, of the instantaneous change in network state $(\frac{dx}{dt} = x(t+1) - x(t))$, and its input and recurrent subspaces projections, respectively (*Equation 4*). Similarly, the total distance traversed in each subspace was calculated as the sum of the magnitude of instantaneous change in network state in the respective subspace, over the duration of the warped utterance.

## Analysis of parallel trajectories

In *Figure 8—figure supplements 1–2*, we present evidence in support of parallel trajectories produced by the network dynamics in response to warped utterances. For each digit, the trajectories were first temporally aligned to the 100% (1x) warp, by matching the time indices of the respective warped utterances to the 100% reference. The following procedure was then independently applied to the temporally aligned recurrent and input subspace trajectories. One separation vector, $S_{0.7x}(t_{aligned})$ ($S_{1.4x}(t_{aligned})$), was calculated at each aligned time point of the trajectory at the fast 0.7x (slow 1.4x) warp, as the difference between the population state ($x$) of the 0.7x (1.4x) warp and 1x warp trajectories. The separation vectors were then normalized to unit length, denoted as $S_{0.7x}^{norm}(t_{aligned})$ ($S_{1.4x}^{norm}(t_{aligned})$). Finally, the projected separations at each fast (slow) warp were generated by calculating the respective separation vectors and projecting them onto the normalized separation vectors of the 0.7x (1.4x) warp trajectory. For example, projected separations at the 0.5x warp, $PS_{0.5x}(t_{aligned})$, were generated by projecting the separation vectors for the 0.5x warp trajectory, onto $S_{0.7x}^{norm}(t_{aligned})$, to give $PS_{0.5x}(t_{aligned})=S_{0.5x}(t_{aligned}).S_{0.7x}^{norm}(t_{aligned})$. The variance explained by these projections was calculated as the ratio between the total population variance of the projected and overall separation.

## Trajectory visualization

For *Figure 3*, PCA was performed on a concatenation of the trajectories generated by both the reservoir and the trained network in response to all utterances of the digits 'six' and 'eight' by a single subject. Trajectories were then individually projected onto three principal components (PCs) and plotted in 3D. The sensory (motor) trajectories were projected onto PCs 2–4 (PCs 1–3). PC one was not used in plotting the sensory trajectories, because it captured features common to both spoken digits. Similarly, in *Figure 6*, PCA was performed on a concatenation of the external inputs for the template utterance of digit zero, a natural test utterance of digit zero by the same subject, and one artificial test utterance each (shuffled and orthogonal noise) derived from these utterances. The external inputs were then projected on the first three PCs and plotted in 3D. In *Figure 8*, PCA was performed on concatenations of temporally-aligned 100 ms segments (i.e. segments aligned to 200–300 ms of the 1x warp) of the external input for digit one at warps of 0.57x, 0.7x, 0.87x, 1x, 1.15x, 1.4x and 1.74x. The external input segments where then projected onto the first three PCs and plotted in 3D. The same procedure was followed in plotting the PCA projections of population state responses in *Figure 8*, and of the recurrent and input subspace population state in *Figure 8—figure supplements 1–2*.

## Source code

Code for the simulation and training of the RNN model, cochleogram data and a set of trained weight matrices are available at https://github.com/vgoudar/SensoriMotorTranscription (*Goudar and Buonomano, 2018*). A copy is archived at https://github.com/elifesciences-publications/SensoriMotorTranscription.

# Acknowledgements

We thank Nicholas Hardy, Omri Barak, Sean Escola, Alexandre Rivkind and Jonathan Kadmon for helpful discussions, and Dharshan Kumaran for comments on an earlier version of this manuscript.

# Additional information

## Funding

| Funder | Grant reference number | Author |
|---|---|---|
| National Science Foundation | NSF IIS-1420897 | Dean V Buonomano |
| Google | Google Faculty Research Award | Dean V Buonomano |
| National Institutes of Health | MH60163 | Dean V Buonomano |

The funders had no role in study design, data collection and interpretation, or the decision to submit the work for publication.

### Author contributions
Vishwa Goudar, Conceptualization, Data curation, Software, Formal analysis, Writing—original draft, Writing—review and editing; Dean V Buonomano, Conceptualization, Supervision, Funding acquisition, Writing—original draft, Writing—review and editing

### Author ORCIDs
Vishwa Goudar https://orcid.org/0000-0002-6612-3076
Dean V Buonomano http://orcid.org/0000-0002-8528-9231

### Decision letter and Author response
Decision letter https://doi.org/10.7554/eLife.31134.017
Author response https://doi.org/10.7554/eLife.31134.018

## Additional files

### Supplementary files
• Transparent reporting form
DOI: https://doi.org/10.7554/eLife.31134.015

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
