## [Decision Letter]

Thank you for submitting your article "Encoding Sensory and Motor Patterns as Time-Invariant Trajectories in Recurrent Neural Networks" for consideration by *eLife*. Your article has been reviewed by three peer reviewers, one of whom is a member of our Board of Reviewing Editors, and the evaluation has been overseen by Richard Ivry as the Senior Editor. The reviewers have opted to remain anonymous.

The reviewers have discussed the reviews with one another and the Reviewing Editor has drafted this decision to help you prepare a revised submission.

This manuscript reports a computational study in which a recurrent neural network was trained to categorize spatio-temporal input stimuli and produce spatio-temporal patterns as responses. More specifically, the study used spoken digits as input stimuli, and transcriptions of these digits as outputs. The authors show that the trained network is robust to variations in the inputs and to external perturbations. In particular, they show that the network generalizes to untrained utterances of digits, and when trained on utterances of different length is able to deal with temporal stretching and compression of the stimuli. These results are interpreted in terms of stimulus-elicited neural trajectories, which show a degree of temporal invariance.

The paper is clearly a well-described, well-thought out computational experiment in artificial recurrent neural networks. However, all reviewers felt that its biological conclusions remain unclear. On a computational level, we know from much work in machine learning, where the task that the authors consider would be called a sequence-to-sequence task, that recurrent neural networks are indeed able to instantiate such computations (e.g., Sutskever et al., 2014). We also know, as the authors themselves report, that alternative mechanisms for dealing with time warping exist.

Accordingly, just the fact that the trained networks could solve the task is not in and of itself a strong biological finding. There are three ways the reviewers feel the paper could be improved, in order of importance:

1) What is the novel biological insight offered by the study? At present there is little detail on this. One prediction appears in the text: "Specifically, the observation that slower stimuli yield trajectories with larger curvature radii implies that the neural population activity should exhibit larger fluctuations in their firing rates in response to slower stimuli". However, in order for that to be a true prediction one would need to show that this is a necessary, or at least generic property, for instance by studying multiple different types of tasks involving temporal warping, or showing somehow that without this increase in curvature radius the trajectories cannot be learned. Moreover, the effect size for this prediction shown in Figure 8—figure supplement 1 appears quite mild (25% increase in range across the full span of warping). both generally and at the mechanistic level.

2) What is the extent and nature of the generalization that these networks are capable of. The authors demonstrate some interpolation and extrapolation of time-warping as well as an ability to reject certain kinds of noise. A stronger test might be for instance whether a network given a training set with naturally spoken examples all time-warped to the same duration, would still be able to generalize to stimuli of different durations?

3) How generic are the results? Is this approach different from previous generic deep learning results? Can the authors describe better the underlying mechanism, Can they convincingly say for instance that the mechanism that they describe in Figure 8 for example is typical, or necessary? That it would remain the same given different choices about training paradigms, etc.

We realize that these are difficult points to address (especially point three). We feel these will be the best ways to strengthen the biological conclusions of the paper, even if a completely full answer cannot be achieved.

To help you in addressing these three main concerns, please find below more detailed reviewer remarks, culled from the original review. Please don't feel you have to address each of these as separate points, I add them here to help you understand in more detail the three main points to be addressed which we listed above.

Major concerns:

1) The network was trained in two steps in which the recurrent weights were trained and then the output weights were trained. I understand the rationale as it is stated in the main text. My question is whether this two-step training is necessary. In a biological case, the network would likely not be trained in such a way. Training would happen presumably all at once based only on the correctness of the output. It would be interesting to show how the network performs when the recurrent and output weights are trained simultaneously. What are the main differences in functionality the network can achieve? Also, a little more text on the rationale for the training procedure would be helpful to reader in the main text.

2) One of the main novelties of the paper is the temporal invariance of the network. The authors state that other models have been developed that can also account for temporal invariance. However, these other models are not fully introduced, forcing the reader to go to the literature to make a comparison. It would be helpful if the text contained a longer introduction to other models and a discussion of how, in functional terms and experimental predictions, the new model presented here compares to and differs from the published models cited in the text.

3) The paper emphasizes the idea of generalizing to different speeds of stimuli. The current results show that when trained on stimuli of different durations, generalization can occur. However, it is unclear if this generalization is a natural consequence of encoding stimuli as trajectories or requires training on different durations. That is, if the network is given a training set with naturally spoken examples all time-warped to the same duration, would the network be able to generalize to stimuli of different durations? The answer to this question helps the reader better understand the scope of the generalization that can occur.

4) How does the performance of the network differ for things like generalization, robustness to noise, etc. as a function of the size of the network? Different figures seem to use different size networks (n = 2100 in Figure 3 = 4000 in Figure 1), but it was not immediately clear how the network size was chosen.

5) One weakness of the paper is that the extent of mechanistic exploration of the RNN and experimental predictions is rather limited, at least in my opinion. I do not have specific suggestions for how to improve on this point. One possibility is further examination of the recurrent weight matrix to look for specific structure and dynamics that could generate additional experimental predictions. Given that I do not have specific questions here, I do not expect that something must be done in this area. I simply note this point because the impact of the paper will be enhanced if more mechanistic insight into how the RNN functions or into experimental predictions could be added.

6) The relation between this work and the substantial progress in deep learning on general sequence-to-sequence tasks (for example, "Sequence to sequence learning with neural networks" by Sutskever et al., 2014 and the dozens of subsequent papers that built upon it) is unclear. Most of these works in deep learning use backpropagation through time (BPTT) to train their networks, an algorithm which is conceptually very distinct from their tamed chaos method. Given the excellent track record of BPTT on many real-world sequence-to-sequence tasks, it seems that the point that the brain can use recurrent dynamics to solve such tasks is already evident. Perhaps the authors feel their approach is more appropriate in some way? If so, it would be important to compare the performance of the tamed chaos method with that of BPTT on more complex tasks and understand the relative advantages and shortcomings between them (for example, is there a task that one method can easily achieve but the other can't? Or is there some feature of the dynamics/networks that are learnt that appears more reasonable for one than the other). Moreover, given these two largely distinct methods to train RNNs, an important question that needs to be addressed is how the mechanisms by which generalization occurs in each method differ from each other. In other words, it is not clear which aspects of the mechanisms they identified in this paper are unique to the tamed chaos method or common to more general classes of algorithms to train RNNs. This paper provides a rather detailed analysis of how their particular training method enables generalization, but, for their analysis to be more useful in understanding how the RNNs in the brain operate, showing that this is a general mechanism in RNNs would be desired.

7) It was unclear how to interpret the fact that the motor-only-training networks were also robust to temporal warping (Figure 7). In fact, for time warping that is in between trained values, the interpolation regime, as the authors call it the motor-only-training networks were at 100% performance and one of the papers the authors cite (Lerner et al., 2014) shows that there is temporal scaling in human hemodynamic imaging mainly in the behavioral range, that is in the interpolation range. Given the chaotic nature of the networks pre-training wouldn't motor-trained-only networks have initial conditions that are almost as different for within digit than between digit? If this is true and the networks can still learn to be at 100% wouldn't that imply that the sensory training is in some sense redundant? I see the greater separation in Figure 7 but it is hard to evaluate whether the extra separation is important or overkill.

8) In general the treatment of temporal warping somewhat confusing. The text is written as if the solution to a time warped input is necessarily to generate the same trajectory running at a different speed. This doesn't have to be the case, in general all that is needed is that whatever dynamics happen, at the transition between the sensory and motor phases the within digit distance (including time warps, different utterances, etc.) be smaller than the between digit distance. The easiest way for this to be achieved is to arrive at the exact same point for different time warps. However, this is not necessary, nor is it what the network actually ends up doing (Figure 8). It was therefore very difficult for me to interpret the significance of Figure 8, that the network arrives at a mid-way solution between constant speed and constant distance. Is there something particularly interesting about this midpoint? Is there a reason it arrived at this compromise between these two possible mechanisms? Is it influenced by the way the authors generate the target trajectories by scaling?

9) The authors make one main biological prediction: "Specifically, the observation that slower stimuli yield trajectories with larger curvature radii implies that the neural population activity should exhibit larger fluctuations in their firing rates in response to slower stimuli". However, in order for that to be a true prediction at least in my mind one would need to show that this is a necessary, or at least generic property, for instance by studying multiple different types of tasks involving temporal warping, or showing somehow that without this increase in curvature radius the trajectories cannot be learned. Moreover, the effect size for this prediction shown in Figure 8—figure supplement 1 appears quite mild (25% increase in range across the full span of warping).

10) Focusing on a specific set of realistic, complex stimuli and outputs is certainly a clear proof of concept of the task. Yet, the shortcoming of such an approach is that it is not immediately clear which part of the reported results are to some extent general, and what is in contrast specific to the particular implementation of the trained network (which relies on a very specific training procedure, and no doubt involves some fine-tuning). I guess what I am missing is a simplified, computational description of the underlying mechanism. Right now, I am left wondering how much details matter, and what happens when some details are changed. For instance, it seems that the specific implementation used in the paper requires that the motor action starts at a fixed time after the stimulus, and that it could not easily accommodate a variable delay between the stimulus and the motor output (e.g. specified by a go cue). Providing a simplified computational description of the mechanism, and examples of extensions/limitations would greatly improve the paper.

[Editors' note: further revisions were requested prior to acceptance, as described below.]

Thank you for submitting a revision of your article "Encoding Sensory and Motor Patterns as Time-Invariant Trajectories in Recurrent Neural Networks" for consideration by *eLife*.

We found the manuscript improved. In particular, the inclusion of the back propagation trained network results that came up with a similar regime to the intrinsic trajectory training was informative and useful.

However, this addition does not fully address the three summary issues raised in the original decision letter. We recognize that these are very difficult to address, and hence a complete answer may not be possible. However, we think it would be appropriate to be more explicit about the limitations of the current work/approach. As one reviewer noted, the current manuscript presents the problems as more solved than warranted and thought it would be useful to note issues that remain relatively unsolved. Your response letter takes more of an approach along these lines than the Discussion section in the revision. For example, in the biological predictions part, it would be helpful to cleanly separate previously known and relatively generic predictions, such as input driven suppression of variability and generalization being challenging, from those that are more specific to the model such as the larger curvature radii. We think this more tempered approach will make the paper more impactful, helping lay out some issues in need of future exploration

---

## [Author Response]

There are three ways the reviewers feel the paper could be improved, in order of importance:1) What is the novel biological insight offered by the study? At present there is little detail on this. One prediction appears in the text: "Specifically, the observation that slower stimuli yield trajectories with larger curvature radii implies that the neural population activity should exhibit larger fluctuations in their firing rates in response to slower stimuli". However, in order for that to be a true prediction one would need to show that this is a necessary, or at least generic property, for instance by studying multiple different types of tasks involving temporal warping, or showing somehow that without this increase in curvature radius the trajectories cannot be learned. Moreover, the effect size for this prediction shown in Figure 8—figure supplement 1 appears quite mild (25% increase in range across the full span of warping). both generally and at the mechanistic level.

It is of course critical for a computational study to generate, and clearly enunciate, predictions, and we have now clarified what we view as the three most important predictions in the Discussion. Specifically:

i) Neural trajectories will be stable in response to local perturbations—potentially administered through optogenetic stimulation—and importantly that the neural trajectories elicited during sensory stimuli will be more resistant to perturbations than the neural trajectories unfolding during the motor epoch of sensory-motor tasks.

ii) Slower stimuli yield trajectories with larger curvature radii, implying that the neural population activity should exhibit larger fluctuations in their firing rates in response to slower stimuli.

iii) Trained networks fail at generalizing to unfamiliar spectral noise patterns, thereby predicting that trajectories in sensory areas will diverge more when presented with stimuli composed of artificial or unfamiliar spectral noise patterns. This prediction could be tested, for example, by training an animal to recognize sounds with noise injected on some principal components of the stimulus spectrogram but not others, and then tested on noise injected along the held-out PC dimensions.

We should clarify that the model does not predict that “without an increase in curvature radius the trajectories cannot be learned”. But rather that temporally invariant encoding of sensory patterns is a result of temporally scaled neural trajectories, wherein the change in the trajectory curvatures radius underlying this temporal scaling is significantly above zero (variable speed/constant distance) and below the warp factor of the sensory pattern (constant speed/variable distance). The reviewer correctly noticed that Figure 8—figure supplement 1 predicts a change in firing rate of approximately 25% across the range of warps tested. This is because we predict that temporal scaling is a result of both a change in curvature and a change in linear speed (Figure 8), and therefore the expected change in the curvature radius (and consequently the firing rate range) should be less than the warp factor. While the magnitude of the change, may indeed make this prediction harder to test, detecting changes in firing rates of 25% is well within the range of experimental studies. For example the effects of attention on firing rate generally fall, on average, within this range (Gregoriou et al., 2014; Maunsell and Treue, 2006).

2) What is the extent and nature of the generalization that these networks are capable of. The authors demonstrate some interpolation and extrapolation of time-warping as well as an ability to reject certain kinds of noise. A stronger test might be for instance whether a network given a training set with naturally spoken examples all time-warped to the same duration, would still be able to generalize to stimuli of different durations?

This is an important question that is partially addressed in point RC7 (see below) and in the motorepoch only training shown in Figure 7. Specifically, a strength of the RNN approach is that encoding spatiotemporal stimuli in continuous neural trajectories endows it with some intrinsic sensory temporal invariance. Thus, training the network on utterances of the same duration results in good temporal scaling (similar to the M-trained RNN in Figure 7, see also the Reservoir control Figure 8). However, it remains the case that training across different durations significantly improves interpolation and extrapolation (as shown in Figure 7). We have now expanded our discussion of the ability of RNNs to exhibit intrinsic temporal scaling.

3) How generic are the results? Is this approach different from previous generic deep learning results? Can the authors describe better the underlying mechanism, Can they convincingly say for instance that the mechanism that they describe in Figure 8 for example is typical, or necessary? That it would remain the same given different choices about training paradigms, etc.

This point, raised in more detail under point RC6 below, relates to determining whether our results and predictions are general or specific to the learning rule we used. To test whether our results depend on the “innate” learning framework, we trained a recurrent network, similar to the ones used in our study, on the sensorimotor task with backpropagation through time (BPTT) applied via ADAM optimization (Kingma and Ba, 2014), a common training approach in the deep learning literature. Training and testing the network with temporally warped utterances (as in Figure 8 and Figure 8—figure supplement 1) yielded qualitatively similar results. Specifically, temporal scaling was observed across a wide range of speeds, however, performance was weaker than that of our innate training approach (compare Figure 8—figure supplement 2 with Figure 7). As stressed in RC6 below, this result strengthens our prediction relating to the mechanisms underlying temporal scaling.

Regarding the comparison with standard deep-learning approaches we would argue that there are there are fundamental differences. Specifically, standard deep-learning approaches generally do not operate in continuous time nor address the issue of dynamic/transient attractors and chaos. For example, the seminal 2014 paper by Sutskever et al. dealt with translation of sentences from one language to another, the inputs in their case were not auditory and time is non-continuous— as a result, temporal invariance was not addressed in their work. But deep learning has, of course, been extensively applied to the domain of artificial speech recognition (Chan et al., 2016; Graves et al., 2006), either using standard feedforward CNN networks or RNNs. However, the RNN models have focused primarily on LSTM networks, and thus their architecture is farther removed from biological realism:

* LSTM units possess state-dependent input and output gates, effectively yielding statedependent, time-varying and arbitrary effective time-constants per cell. As a result, temporal scaling within these models may result from LSTM gating dynamics, wherein the effective time-constants of the cells are warp-dependent. However, to the best of our knowledge, neurons or neural populations with this state-dependent and time-varying time constants have not been experimentally observed.

* Deep learning models often enlist bidirectional LSTMs to allow for current inputs to be processed in light of past *and future* context. While this serves to improve efficiency, it is certainly not biologically plausible.

Major concerns:1) The network was trained in two steps in which the recurrent weights were trained and then the output weights were trained. I understand the rationale as it is stated in the main text. My question is whether this two-step training is necessary. In a biological case, the network would likely not be trained in such a way. Training would happen presumably all at once based only on the correctness of the output. It would be interesting to show how the network performs when the recurrent and output weights are trained simultaneously. What are the main differences in functionality the network can achieve? Also, a little more text on the rationale for the training procedure would be helpful to reader in the main text.

This is a good question, although it is not clear if separate stages of training in sensory-motor tasks should be considered less biological. For example, in sensory-motor learning in the song bird, there are clear distinctions between learning of the sensory component (the template) and motor learning—indeed the same is probably true in speech (Doupe and Kuhl, 1999). Nevertheless, it is not the case that learning has to be performed in two steps. Indeed one could interpret our training framework as having parallel plasticity, with learning rates of the recurrent weights being much larger than that of the output weights. Furthermore, we now establish that end-to-end training of the network with BPTT does not alter the mechanism underlying temporal invariance. We have now added text elaborating on this point.

2) One of the main novelties of the paper is the temporal invariance of the network. The authors state that other models have been developed that can also account for temporal invariance. However, these other models are not fully introduced, forcing the reader to go to the literature to make a comparison. It would be helpful if the text contained a longer introduction to other models and a discussion of how, in functional terms and experimental predictions, the new model presented here compares to and differs from the published models cited in the text.

We thank the reviewer for this suggestion. We have now updated the Discussion section with a more detailed descriptions of other models. Specifically, the main other biologically realistic model for temporally invariant processing of sensory patterns proposes synaptic shunting as a mechanism that amounts to a modulation of the time constants of the neurons by the warped input—in other words it is more akin to a cellular mechanism than a network mechanism (Gutig and Sompolinsky, 2009).

3) The paper emphasizes the idea of generalizing to different speeds of stimuli. The current results show that when trained on stimuli of different durations, generalization can occur. However, it is unclear if this generalization is a natural consequence of encoding stimuli as trajectories or requires training on different durations. That is, if the network is given a training set with naturally spoken examples all time-warped to the same duration, would the network be able to generalize to stimuli of different durations? The answer to this question helps the reader better understand the scope of the generalization that can occur.

Please see response #2 above.

4) How does the performance of the network differ for things like generalization, robustness to noise, etc. as a function of the size of the network? Different figures seem to use different size networks (n = 2100 in Figure 3 = 4000 in Figure 1), but it was not immediately clear how the network size was chosen.

The size of the network certainly affects its capacity, particularly for generalization. The reason for using a larger network in Figure 1 was to address cross-speaker generalization. The smaller network size used in the other figures was not as effective when trained and tested across speakers. We now explain this in Materials and methods section of the paper.

5) One weakness of the paper is that the extent of mechanistic exploration of the RNN and experimental predictions is rather limited, at least in my opinion. I do not have specific suggestions for how to improve on this point. One possibility is further examination of the recurrent weight matrix to look for specific structure and dynamics that could generate additional experimental predictions. Given that I do not have specific questions here, I do not expect that something must be done in this area. I simply note this point because the impact of the paper will be enhanced if more mechanistic insight into how the RNN functions or into experimental predictions could be added.

We share the reviewer’s feelings here, and we think the point reflects the deeper question of what it means to “understand” an RNN. Indeed, this was a concern raised in the first review, and we believe the mechanistic analyses including the decomposition analysis that was added to the first revision is a strong approach—but admittedly the field as a whole must address the larger issue at stake.

6) The relation between this work and the substantial progress in deep learning on general sequence-to-sequence tasks (for example, "Sequence to sequence learning with neural networks" by Sutskever et al., 2014 and the dozens of subsequent papers that built upon it) is unclear. Most of these works in deep learning use backpropagation through time (BPTT) to train their networks, an algorithm which is conceptually very distinct from their tamed chaos method. Given the excellent track record of BPTT on many real-world sequence-to-sequence tasks, it seems that the point that the brain can use recurrent dynamics to solve such tasks is already evident. Perhaps the authors feel their approach is more appropriate in some way? If so, it would be important to compare the performance of the tamed chaos method with that of BPTT on more complex tasks and understand the relative advantages and shortcomings between them (for example, is there a task that one method can easily achieve but the other can't? Or is there some feature of the dynamics/networks that are learnt that appears more reasonable for one than the other). Moreover, given these two largely distinct methods to train RNNs, an important question that needs to be addressed is how the mechanisms by which generalization occurs in each method differ from each other. In other words, it is not clear which aspects of the mechanisms they identified in this paper are unique to the tamed chaos method or common to more general classes of algorithms to train RNNs. This paper provides a rather detailed analysis of how their particular training method enables generalization, but, for their analysis to be more useful in understanding how the RNNs in the brain operate, showing that this is a general mechanism in RNNs would be desired.

This is a very important point. As mentioned above (Summary point #3), the differences between the two techniques are more than just the training rule. Specifically, in regard to the treatment of time. E.g., in Sutskever *et al. 2014,* time is not explicitly present, and in many of the other models the temporal structure of the input is “spatialized” (i.e., at time step t discrete time step inputs from t-n, t-n-1, …, t, are presented to the network)—and thus does not capture how the brain represents time. Nevertheless to address the reviewers question we have now trained our network using a version of BPTT. Under some conditions (random orthogonal initialization of weight matrices with a high gain) BPTT resulted in good temporal scaling via similar mechanisms (temporal scaling of neural trajectories by changes in curvature radius)—but overall performance, as measured by CNN “handwritten” classification, was inferior. These results establish that the same solution was arrived at using different training methods, implying that our predictions are fairly general (even if some rules result in better performance than others). This result is consistent with our stance that these models are meant to capture computational principles of biological of recurrent neural networks, but the learning rules by which biological circuits reach these regimes remain to be elucidated.

These results are presented in Figure 8—figure supplement 2.

7) It was unclear how to interpret the fact that the motor-only-training networks were also robust to temporal warping (Figure 7). In fact, for time warping that is in between trained values, the interpolation regime, as the authors call it the motor-only-training networks were at 100% performance and one of the papers the authors cite (Lerner et al. 2014) shows that there is temporal scaling in human hemodynamic imaging mainly in the behavioral range, that is in the interpolation range. Given the chaotic nature of the networks pre-training wouldn't motor-trained-only networks have initial conditions that are almost as different for within digit than between digit? If this is true and the networks can still learn to be at 100% wouldn't that imply that the sensory training is in some sense redundant? I see the greater separation in Figure 7 but it is hard to evaluate whether the extra separation is important or overkill.

The network is initialized to a chaotic state which means it is chaotic in the absence of any external input, but as has been described, an external input can suppress this chaotic activity (Rajan et al., 2010). A nice feature of our RNN approach is that it does show some degree of intrinsic temporal scaling because the external input can partially “clamp” the internal dynamics (of course, sensory-epoch training leads to substantial performance improvements in terms of generalization to spectral noise and to different speakers (Figure 1) as well as improved temporal scaling (Figure 7)). This was discussed in the Results (*Stability of the Neural Trajectories*), and we now expand upon it in the Discussion section.

8) In general the treatment of temporal warping somewhat confusing. The text is written as if the solution to a time warped input is necessarily to generate the same trajectory running at a different speed. This doesn't have to be the case, in general all that is needed is that whatever dynamics happen, at the transition between the sensory and motor phases the within digit distance (including time warps, different utterances, etc.) be smaller than the between digit distance. The easiest way for this to be achieved is to arrive at the exact same point for different time warps. However, this is not necessary, nor is it what the network actually ends up doing (Figure 8). It was therefore very difficult for me to interpret the significance of Figure 8, that the network arrives at a mid-way solution between constant speed and constant distance. Is there something particularly interesting about this midpoint? Is there a reason it arrived at this compromise between these two possible mechanisms? Is it influenced by the way the authors generate the target trajectories by scaling?

This is a fascinating question regarding the nature of the network’s dynamics. The reviewer is certainly correct that other solutions could be used during the sensory-epoch. But it is more complicated than requiring that “at the transition between the sensory and motor phases the within digit distance (including time warps, different utterances, etc.) be smaller than the between digit distance”. Specifically, there is an interaction between this distance and the basin of attraction of the motor dynamic attractor: it is possible that at the end of the sensory epoch, despite being closer to the other within-digit trajectories, a trajectory may not end up within the basin of attraction of the corresponding digit’s motor dynamic attractor. Good performance will depend on the relationship between the size of this basin and the within digit-distance at the end of the sensory epoch. In other words, the size of the basin of attraction poses a stronger constraint on the withindigit distance. Since both of these characteristics of the network’s dynamics are affected by training, it appears that the network finds the best tradeoff. We now further highlight this interaction.

9) The authors make one main biological prediction: "Specifically, the observation that slower stimuli yield trajectories with larger curvature radii implies that the neural population activity should exhibit larger fluctuations in their firing rates in response to slower stimuli". However, in order for that to be a true prediction at least in my mind one would need to show that this is a necessary, or at least generic property, for instance by studying multiple different types of tasks involving temporal warping, or showing somehow that without this increase in curvature radius the trajectories cannot be learned. Moreover, the effect size for this prediction shown in Figure 8—figure supplement 1 appears quite mild (25% increase in range across the full span of warping).

Please see the response to Summary point #1.

10) Focusing on a specific set of realistic, complex stimuli and outputs is certainly a clear proof of concept of the task. Yet, the shortcoming of such an approach is that it is not immediately clear which part of the reported results are to some extent general, and what is in contrast specific to the particular implementation of the trained network (which relies on a very specific training procedure, and no doubt involves some fine-tuning). I guess what I am missing is a simplified, computational description of the underlying mechanism. Right now, I am left wondering how much details matter, and what happens when some details are changed. For instance, it seems that the specific implementation used in the paper requires that the motor action starts at a fixed time after the stimulus, and that it could not easily accommodate a variable delay between the stimulus and the motor output (e.g. specified by a go cue). Providing a simplified computational description of the mechanism, and examples of extensions/limitations would greatly improve the paper.

We thank the reviewer for this comment. The addition of a model using a different learning rule addresses, to some degree, the question of generality (Figure 8—figure supplement 2). But the reviewer is correct that in its current implementation the model does not adapt well to a variable sensory-motor delay. Arguably, in biological systems such delayed output invokes additional working memory dynamics, which could be incorporated by training the network to converge to a fixed-point attractor before the go signal. Regarding the presentation of a reduced model, this is something we have struggled with. Specifically, it is challenging to present a reduced model of something that we view as a truly emergent phenomenon (indeed, emergent phenomena can be defined as those that are not easily reducible). Thus, this relates to point RC5: what does it mean to understand the computations that emerge from a RNN. In our view many of the complex computations the brain performs are indeed emergent phenomena, and as such the field will need to address the question of how to study, perturb, establish causality, and understand these networks. Hopefully, this paper takes an initial step in this direction, by highlighting the challenge, showing that RNNs can solve complex sensori-motor tasks, and offering a tool to understand these dynamics (the RNN decomposition).

[Editors' note: further revisions were requested prior to acceptance, as described below.]

Thank you for submitting a revision of your article "Encoding Sensory and Motor Patterns as Time-Invariant Trajectories in Recurrent Neural Networks" for consideration by eLife.We found the manuscript improved. In particular, the inclusion of the back propagation trained network results that came up with a similar regime to the intrinsic trajectory training was informative and useful.However, this addition does not fully address the three summary issues raised in the original decision letter. We recognize that these are very difficult to address, and hence a complete answer may not be possible. However, we think it would be appropriate to be more explicit about the limitations of the current work/approach. As one reviewer noted, the current manuscript presents the problems as more solved than warranted and thought it would be useful to note issues that remain relatively unsolved. Your response letter takes more of an approach along these lines than the Discussion section in the revision. For example, in the biological predictions part, it would be helpful to cleanly separate previously known and relatively generic predictions, such as input driven suppression of variability and generalization being challenging, from those that are more specific to the model such as the larger curvature radii. We think this more tempered approach will make the paper more impactful, helping lay out some issues in need of future exploration

We thank the reviewers for their further feedback, and for raising the point pertaining to making explicit statements about the limitations of the model, and issues that remain unaddressed. We, of course, did not mean to imply in the text that questions relating to how recurrent neural networks perform sensorimotor computations are solved, and hopefully made that clear in the letter, if not in the Discussion section. We have now added an additional paragraph to the Discussion section about limitations and future directions (see below). Finally, we have also clarified the predictions of the model with the goal of helping the reader understand how specific they are likely to be to the current model.

“These experimental predictions are critical to validate the model’s implementation of complex and invariant sensorimotor computations as stable neural trajectories. However, even if validated a number questions remain to be addressed. Most notably, how can the recurrent synaptic strengths be tuned to develop stable trajectories in a biologically plausible manner? The learning rule used here, coupled with its requirement of an explicit target for each recurrent unit, make it biologically implausible. However, while it is important that the sensorimotor representations are encoded as locally stable trajectories, the structure of the target trajectories themselves are essentially arbitrary. It is therefore conceivable that arbitrary yet locally-stable encoding trajectories may emerge from unsupervised learning. A second related issue is that to achieve strong temporal invariance, our model had to be trained over a range of sensory stimuli speeds. Again, it is not clear if this would represent a biologically plausible scenario—to help address this question, it will be important for future research to determine if the ability to recognize stimuli independent of speed is learned through experience. Finally, questions relating to the learning capacity of networks capable of strong temporal invariance, and the expedience of sensory-epoch training for temporal invariance remain open.”